# Coronary arterial development is regulated by a Dll4-Jag1-EphrinB2 signaling cascade

Stanislao Igor Travisano[1,2], Vera Lucia Oliveira[1,2], Belén Prados[1,2], Joaquim Grego-Bessa[1,2], Rebeca Piñeiro-Sabarís[1,2], Vanesa Bou[1,2], Manuel J Gómez[3], Fátima Sánchez-Cabo[3], Donal MacGrogan[1,2]*, José Luis de la Pompa[1,2]*

[1]Intercellular Signalling in Cardiovascular Development and Disease Laboratory, Centro Nacional de Investigaciones Cardiovasculares Carlos III (CNIC), Madrid, Spain; [2]CIBER de Enfermedades Cardiovasculares, Madrid, Spain; [3]Bioinformatics Unit, Centro Nacional de Investigaciones Cardiovasculares, Madrid, Spain

**Abstract** Coronaries are essential for myocardial growth and heart function. Notch is crucial for mouse embryonic angiogenesis, but its role in coronary development remains uncertain. We show Jag1, Dll4 and activated Notch1 receptor expression in sinus venosus (SV) endocardium. Endocardial *Jag1* removal blocks SV capillary sprouting, while *Dll4* inactivation stimulates excessive capillary growth, suggesting that ligand antagonism regulates coronary primary plexus formation. Later endothelial ligand removal, or forced expression of Dll4 or the glycosyltransferase Mfng, blocks coronary plexus remodeling, arterial differentiation, and perivascular cell maturation. Endocardial deletion of *Efnb2* phenocopies the coronary arterial defects of Notch mutants. Angiogenic rescue experiments in ventricular explants, or in primary human endothelial cells, indicate that EphrinB2 is a critical effector of antagonistic Dll4 and Jag1 functions in arterial morphogenesis. Thus, coronary arterial precursors are specified in the SV prior to primary coronary plexus formation and subsequent arterial differentiation depends on a Dll4-Jag1-EphrinB2 signaling cascade.

*For correspondence:
dmacgrogan@cnic.es (DMG);
jlpompa@cnic.es (JLP)

**Competing interests:** The authors declare that no competing interests exist.

## Introduction

Coronary artery disease leading to cardiac muscle ischemia is the major cause of morbidity and death worldwide (*Sanchis-Gomar et al., 2016*). Deciphering the molecular pathways driving progenitor cell deployment during coronary angiogenesis could inspire cell-based solutions for revascularization following ischemic heart disease. The coronary endothelium in mouse derives from at least two complementary progenitor sources (*Tian et al., 2015*) that may share a common developmental origin (*Zhang et al., 2016a*; *Zhang et al., 2016b*). The sinus venosus (SV) commits progenitors to arteries and veins of the outer myocardial wall (*Red-Horse et al., 2010*; *Tian et al., 2013*), and the endocardium contributes to arteries of the inner myocardial wall and septum (*Red-Horse et al., 2010*; *Wu et al., 2012*; *Tian et al., 2013*). Regardless of origin, the endothelial precursors invest the myocardial wall along stereotyped routes and eventually interlink in a highly coordinated fashion. Subsequently, discrete components of the primitive plexus are remodeled into arteries and veins, and stabilized through mural cell investment and smooth muscle cell differentiation (*Udan et al., 2013*). Arterial-venous specification of endothelial progenitors is genetically pre-determined (*Swift and Weinstein, 2009*), whereas arterial differentiation and patterning depend on environmental cues, such as blood flow and hypoxia-dependent proangiogenic signals (*le Noble et al., 2005*; *Jones et al., 2006*; *Fish and Wythe, 2015*). Vascular endothelial growth factor (VEGF) binds

endothelial receptors and drives the expansion of the blood vessel network as a response to hypoxia (*Dor et al., 2001*; *Liao and Johnson, 2007*; *Krock et al., 2011*).

The Notch signaling pathway is involved in angiogenesis in the mouse embryo and in the post-natal retina. In this processes, the ligand Dll4 is upregulated by VEGF, leading to Notch activation in adjacent endothelial cells (ECs), vessel growth attenuation, and maintenance of vascular integrity (*Blanco and Gerhardt, 2013*). In contrast, the ligand Jag1 has a proangiogenic Dll4-Notch–inhibitory function, suggesting that the overall response of ECs to VEGF is mediated by the opposing roles of Dll4 and Jag1 (*Benedito et al., 2009*). Dll4-Notch1 signaling is strengthened in the presence of the glycosyltransferase Mfng (*Benedito et al., 2009*; *D'Amato et al., 2016*).

Our understanding of how coronary vessels originate, are patterned, and integrate with the systemic circulation to become functional is still limited. Coronary arteries are distinct from peripheral arteries in that they originate from SV and ventricular endocardium, which is a specialized endothelium that lines the myocardium. Moreover, SV endothelium has a venous identity, as opposed to the retina vascular bed, which has no pre-determined venous identity. Given these differences of developmental context, it is essential to evaluate the role of Notch in coronary arterial development, and importantly, the implications for heart development and repair.

Several components of the Notch pathway have been examined in the context of coronary artery formation. Inactivation of the Notch modifier *Pofut1* results in excessive coronary angiogenic cell proliferation and plexus formation (*Wang et al., 2017*), while endothelial inactivation of *Adam10*, required for Notch signaling activation, leads to defective coronary arterial differentiation (*Farber et al., 2019*). Transcriptomics has shown that pre-artery cells appear in the immature coronary vessel plexus before coronary blood flow onset, and express Notch genes, including *Dll4* (*Su et al., 2018*). Here, we examine the early and late requirements of Notch ligands Jag1 and Dll4, and their downstream effector EphrinB2, for coronary arterial development.

## Results

### Jag1 and Dll4 are expressed in SV endocardium and coronary vessels endothelium

We examined SV and coronary vessels for the expression of Jag1 and Dll4. At embryonic day 11.5 (E11.5), Jag1 was detected in SV ECs and in ECs extending into the right atrium (*Figure 1A*). Dll4 was also expressed in ECs emanating from the SV and in the endocardium lining the right atrium (*Figure 1A*). These results suggest that either ligand could potentially activate Notch1 in the SV endocardium (*Figure 1A*). At E12.5, N1ICD was detected in endomucin (Emcn)-positive ECs in subepicardial capillaries emerging from the SV (*Figure 1A*). At E13.5, Jag1 and *Dll4*, were expressed in ECs of the developing coronary arteries (intramyocardial vessels; *Figure 1—figure supplement 1A*). *Dll4* and *Mfng* were also expressed in prospective veins (subepicardial vessels; *Figure 1—figure supplement 1A*). At E15.5, Jag1, *Dll4*, and *Mfng* were all expressed in arterial ECs, whereas *Mfng* was still found in subepicardial vessels (*Figure 1—figure supplement 1B*). Thus, Jag1 and Dll4 expression is found in discrete ECs in SV endothelium, abates in sub-epicardial veins and becomes restricted to intramyocardial coronary arteries at later developmental stages.

### Nfatc1-positive progenitors give rise to the majority of subepicardial vessels

To inactivate Notch ligands in SV progenitors we used the *Nfatc1-Cre* driver line (*Wu et al., 2012*). To confirm the SV and endocardial specificity of this line, we crossed it with the *Rosa26-LacZ* reporter line (*Soriano, 1999*). X-gal-staining of heart sections of E11.5 embryos identified patchy LacZ expression in SV endothelium (*Figure 1—figure supplement 2A*). ß-gal staining was consistent with Nfatc1 protein nuclear localization in a subset of ECs lining the SV (*Figure 1—figure supplement 2A*). Uniform ß-gal staining was detected in ventricular endocardium and cushion mesenchyme derived from endocardial cells (*Figure 1—figure supplement 2A*). At E12.5, co-labelling with an anti-Pecam1 antibody revealed ß-gal-positive staining in 60% of Pecam1-positive subepicardial vessels in the right ventricle and about 50% in the left ventricle (*Figure 1—figure supplement 2B—Source data 1*, sheet 1). Tracking the expression of *Nfatc1-Cre*-driven red fluorescent protein (RFP) and the endothelial-specific nuclear protein Erg at E12.5 indicated that about 80% of nuclei in the

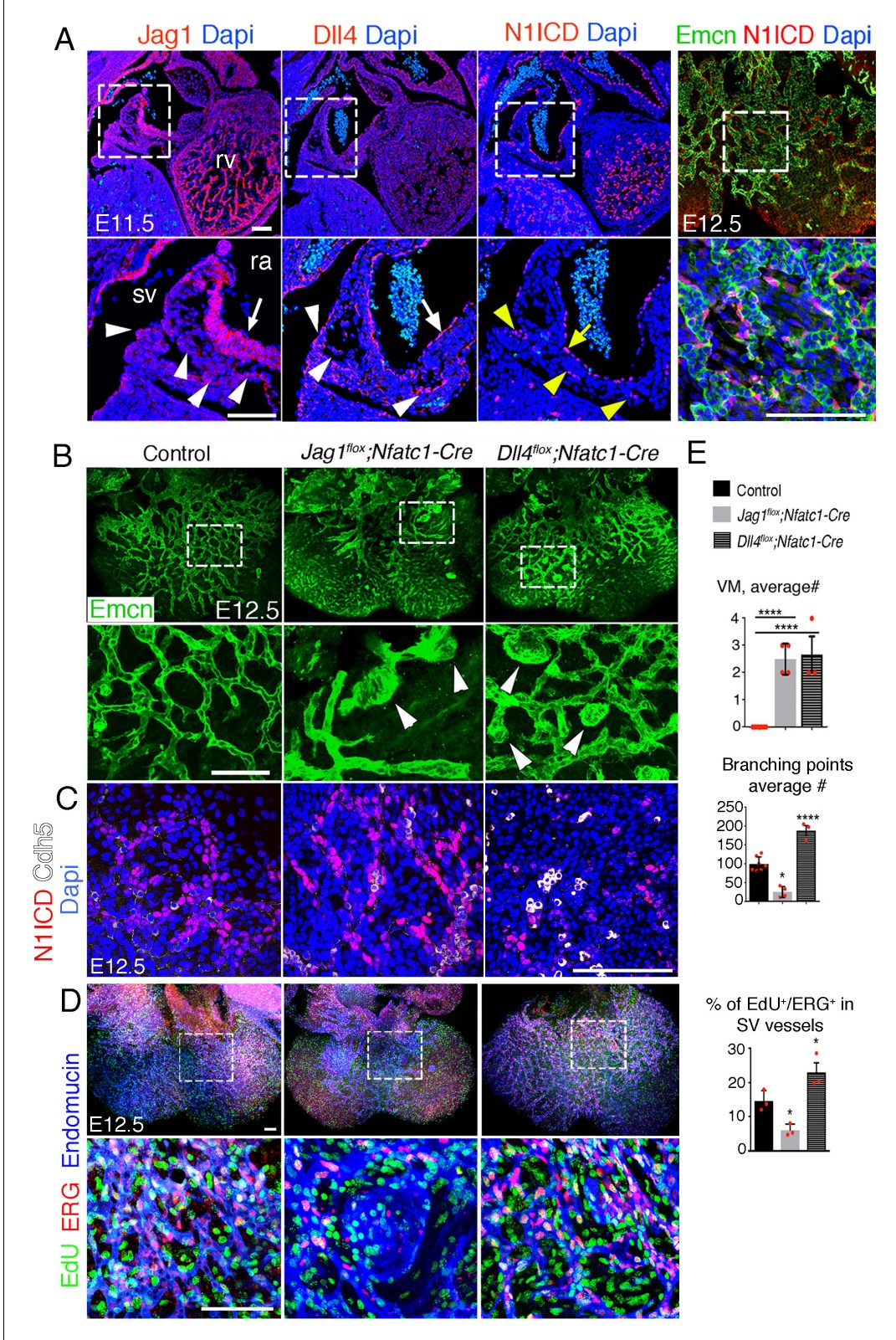

**Figure 1.** Endocardial *Jag1* or *Dll4* inactivation disrupts coronary plexus formation. (**A**) Jag1, Dll4, and N1ICD immunostaining (red) in E11.5 control hearts, sagittal views. Magnified views of boxed areas show details of sinus venosus (sv, arrowheads) and right atrium (ra, arrow). Whole-mount dorsal view of immunostainings for N1ICD (red) and Emcn (green) in E12.5 control heart. Magnified views show detail of sub-epicardial endothelium. Nuclei are counterstained with Dapi (blue). (**B**) Whole-mount dorsal view of immunostaining for Emcn (green) in E12.5 control, *Jag1flox;Nfatc1-Cre*, and *Dll4flox*;

*Figure 1 continued on next page*

Figure 1 continued

*Nfatc1-Cre* mutant hearts. Arrowheads indicate vascular malformations. Quantified data of average number of vascular malformations (VM) and average number of branching points in E12.5 control, *Jag1^flox^;Nfatc1-Cre* and *Dll4^flox^;Nfatc1-Cre* hearts. (C) Dorsal views of whole-mount E12.5 control, *Jag1^flox^; Nfat-Cre*, and *Dll4^flox^;Nfatc1-Cre* hearts stained for N1ICD (red) and VE-Caderin (white). Microscope: Leica SP5. Software: LAS-AF 2.7.3. build 9723. Objective: HCX PL APO CS 10 × 0.4 dry. HCX PL APO lambda blue 20 × 0.7 multi-immersion. (D) Dorsal views of whole-mount E12.5 control, *Jag1^flox^; Nfat-Cre*, and *Dll4^flox^;Nfatc1-Cre* hearts stained for EdU (green), ERG (red), and Emcn (blue). Scale bars, 100 μm. Microscope: Nikon A1-R. Software: NIS Elements AR 4.30.02. Build 1053 LO, 64 bits. Objectives: Plan Apo VC 20x/0.75 DIC N2 dry; Plan Fluor 40x/1.3 Oil DIC H N2 Oil. (E) Quantified data for vascular malformations (VM), average number (#) of branching points and EdU-ERG dual-positive nuclei as a percentage of all nuclei in sub-epicardial vessels. Data are mean ± s.d. (n = 7 control embryos and n = 4 *Jag1^flox^;Nfat-Cre* and n = 3 *Dll4^flox^;Nfatc1-Cre* mutant embryos for VM and average # of branching points. n = 3 control embryos and n = 3 mutant embryos for EdU-ERG). *p<0.05, ****p<0.0001 by one-way ANOVA with Tukey's multiple comparison tests). Abbreviations: rv, right ventricle.

The online version of this article includes the following figure supplement(s) for figure 1:

**Figure supplement 1.** *Jag1*, *Dll4*, and *Mfng* expression in developing coronary vessels.
**Figure supplement 2.** Most ventricular free-wall coronary vessels derive from Nfatc1⁺ progenitors.
**Figure supplement 3.** Coronary vessels of Notch pathway and Notch effector mutants display vascular malformations.
**Figure supplement 4.** Histological and molecular marker analysis of E12.5 *Jag1^flox^;Nfatc1-Cre*, and *Dll4^flox^;Nfatc1-Cre* and E16.5 *Efnb2^flox^;Nfatc1-Cre* hearts.

endothelial network were Nfatc1-positive and Erg-positive (*Figure 1—figure supplement 2B—Source data 1*, sheet 1). Thus, SV-derived Nfatc1-positive progenitors give rise to 50–80% of subepicardial vessels in the ventricular wall, consistent with previous reports (*Chen et al., 2014*; *Cavallero et al., 2015*; *Zhang et al., 2016a*).

To trace the fate of Nfatc1-positive cells relative to Notch activity, we crossed *Nfatc1-Cre;Rosa26-RFP* mice with the Notch reporter line *CBF:H2B-Venus*. At E11.5, a subset of nuclear-stained RFP ECs extending from the SV into the right ventricle were co-labelled with CBF:H2B-Venus (*Figure 1—figure supplement 2D*). At E12.5, some RFP-labelled capillaries on the dorsal side of the heart were co-labeled with CBF:H2B-Venus while others were labelled with CBF:H2B-Venus alone (*Figure 1—figure supplement 2E*), indicating that Notch signaling activity is present in both Nfatc1-positive and Nfatc1-negative populations of SV-derived ECs.

## Opposing roles of *Jag1* and *Dll4* in coronary plexus formation from SV

*Jag1* inactivation with the *Nfatc1-Cre* driver line, specific of SV and endocardium, resulted in death at E13.5 (*Supplementary file 1*). Whole-mount Endomucin (Emcn) staining at E12.5 revealed a well-formed vascular network covering the dorsal aspect of control hearts (*Figure 1B, E—figure supplement 3A—Source data 1*, sheet 2), whereas the vascular network in *Jag1^flox^;Nfatc1-Cre* mutants was poorly developed, and exhibited numerous capillary malformations (*Figure 1B, E—figure supplement 3B—Source data 1*, sheet 2) with decreased endothelial branching (*Figure 1B,E—Source data 1*, sheet 2). N1ICD expression was also more prominent (*Figure 1C*) and EC proliferation reduced 60% relative to control (*Figure 1D,E—Source data 1*, sheet 2). Ventricular wall thickness in endocardial *Jag1* mutants was reduced by 50–60% relative to controls (*Figure 1—figure supplement 4A—Source data 1*, sheet 3). Thus, endocardial *Jag1* deletion causes the growth arrest of the primitive coronary plexus.

Deletion of *Dll4* with the *Nfatc1-Cre* driver resulted in the death of high proportion of embryos at E10.5 (*Supplementary file 1*). Nonetheless, about a third of mutant embryos survived until E12.5 (*Supplementary file 1*). Whole-mount Emcn immunostaining of E12.5 *Dll4^flox^;Nfatc1-Cre* hearts revealed a comparatively denser capillary network (*Figure 1B,E—Source data 1*, sheet 2), characterized by numerous capillary malformations (*Figure 1B* and *Figure 1—figure supplement 3A,C—Source data 1*, sheet 2), and increased endothelial branching (*Figure 1B,E—Source data 1*, sheet 2). N1ICD expression was reduced (*Figure 1C*), as expected, and EC proliferation increased 30% relative to control (*Figure 1D,E—Source data 1*, sheet 2). These defects were associated with 50–60% reduction in ventricular wall thickness (*Figure 1—figure supplement 4A—Source data 1*, sheet 3). Thus, endocardial inactivation of *Dll4* causes excessive growth of the primitive coronary plexus.

## Endocardial *Jag1* or *Dll4* deletion results in a hypoxic and metabolic stress response

To determine the effect of early endocardial *Jag1* or *Dll4* inactivation on cardiac development we performed RNA-seq. This analysis yielded 211 differentially expressed genes (DEG) in the *Jag1^{flox}*; *Nfatc1-Cre* transcriptome (130 upregulated, 81 downregulated; *Figure 2A—Supplementary file 2*) and 274 DEGs in the *Dll4^{flox}*;*Nfatc1-Cre* transcriptome (180 upregulated, 94 downregulated; *Figure 2A—Supplementary file 2*).

Ingenuity Pathway Analysis (IPA) identified enrichment of EC functions (*Figure 2A*, left plot, *Supplementary file 3*). The main terms overrepresented in both genotypes were angiogenesis and EC development, and capillary vessel density, possibly reflecting the lack of a normal-sized capillary network (*Figure 2A—Supplementary file 3*). EC proliferation and heart contraction were predicted to be upregulated in *Jag1^{flox}*;*Nfatc1-Cre* embryos, while EC migration was upregulated in *Dll4^{flox}*; *Nfatc1-Cre* mice (*Figure 2A—Supplementary file 3*). Analysis of upstream regulators revealed activation of hypoxia (Hif1α), acute inflammatory response (Nf-κb1α), intracellular stress pathways (Atf4), and response to metabolic stress pathways (Foxo), while cell cycle and DNA repair pathways (Myc, Tp53) were negatively regulated (*Figure 2A*, right plot).

In situ hybridization (ISH) showed reduced expression of *HeyL* and the Notch target *Efnb2* in sub-epicardial vessels (*Figure 2B*). *Fabp4*, a member of the fatty-acid-binding protein family, was upregulated (*Figure 2A*). Fabp4 is a DLL4-NOTCH target downstream of VEGF and FOXO1 in human EC (*Harjes et al., 2014*), required for EC growth and branching (*Elmasri et al., 2009*). *Fapb4* expression was found exclusively in the atrio-ventricular groove in *Jag1^{flox}*;*Nfatc1-Cre* hearts (*Figure 2B*), but extended sub-epicardially into the base of the heart in *Dll4^{flox}*;*Nfatc1-Cre* hearts (*Figure 2B*). *Vegfa* expression was not globally affected in mutant hearts (*Figure 2B*), despite being upregulated in the RNA-seq (*Figure 2A*). Cell cycle-associated genes such as *Cdkn1b*/p27, a negative regulator of cell proliferation, were also upregulated in both genotypes (*Figure 2A*), and p27 nuclear staining was increased twofold in compact myocardium (*Figure 2C,D—Source data 1*, sheet 4), indicating decreased cellular proliferation. *Connexin 40* (*Cx40*) and *Hey2*, which label trabecular and compact myocardium respectively, showed no alteration in their expression domains by ISH (*Figure 2B*; *Figure 1—figure supplement 4B*), suggesting that chamber patterning was normal in these mutants. Likewise, the glycolytic marker genes *Ldha* and *Pdk1* (*Menendez-Montes et al., 2016*) were confined, as normal, to the compact myocardium (*Figure 2B*; *Figure 1—figure supplement 4B*), indicating maintenance of ventricular chamber metabolic identity.

We examined E12.5 *Jag1^{flox}*;*Nfatc1-Cre* and *Dll4^{flox}*;*Nfatc1-Cre* mutants for evidence of hypoxia given that the gene signatures in the RNA-seq analysis suggested an ongoing hypoxic/metabolic stress response. The hypoxic response might also explain the defect of ventricular wall growth. We performed immunohistochemical detection of pimonidazole on E12.5 *Jag1^{flox}*;*Nfatc1-Cre* and *Dll4^{flox}*;*Nfatc1-Cre* heart sections (*Figure 2—figure supplement 1*). Pimonidazole hydrochloride (Hypoxyprobe) immunostaining for hypoxic tissues was detected most intensely in cells lying within the ventricular septum and atrio-ventricular (AV) groove in the subepicardial area where the primitive coronary endothelium emerges to cover the myocardium (*Figure 2—figure supplement 1Ai*, arrowheads). Immunostaining for Glut1 (*Slc2a1*), a direct HIF1 target, showed myocardial expression partially overlapping with hypoxyprobe distribution (*Figure 2—figure supplement 1Aii*). There was no Glut1 immunostaining in the subepicardial mesenchyme zone (*Figure 2—figure supplement 1Aii*). We found no difference in the intensities of hydroxyprobe or Glut1 staining in either the ventricular free wall, or in the ventricular septum in *Jag1^{flox}*;*Nfatc1-Cre* and *Dll4^{flox}*;*Nfatc1-Cre* hearts (*Figure 2—figure supplement 1*). There was no obvious change in hydroxyprobe immunostaining of endothelial cells emerging at the AV groove either (*Figure 2—figure supplement 1Ai-Ci*). These results indicate that *Jag1^{flox}*;*Nfatc1-Cre* and *Dll4^{flox}*;*Nfatc1-Cre* mutant hearts are not overtly hypoxic at E12.5, suggesting that the hypoxic/metabolic stress gene signatures may be due to a cell autonomous defect of endocardial/endothelial cells.

## Defective coronary remodeling and maturation in endothelial *Jag1* or *Dll4* mutants

We next examined the requirements of endothelial Jag1 and Dll4 for coronary vessel remodeling and maturation. To circumvent the early lethality of *Jag1^{flox}*- or *Dll4^{flox}*;*Nfatc1-Cre* mutants, we

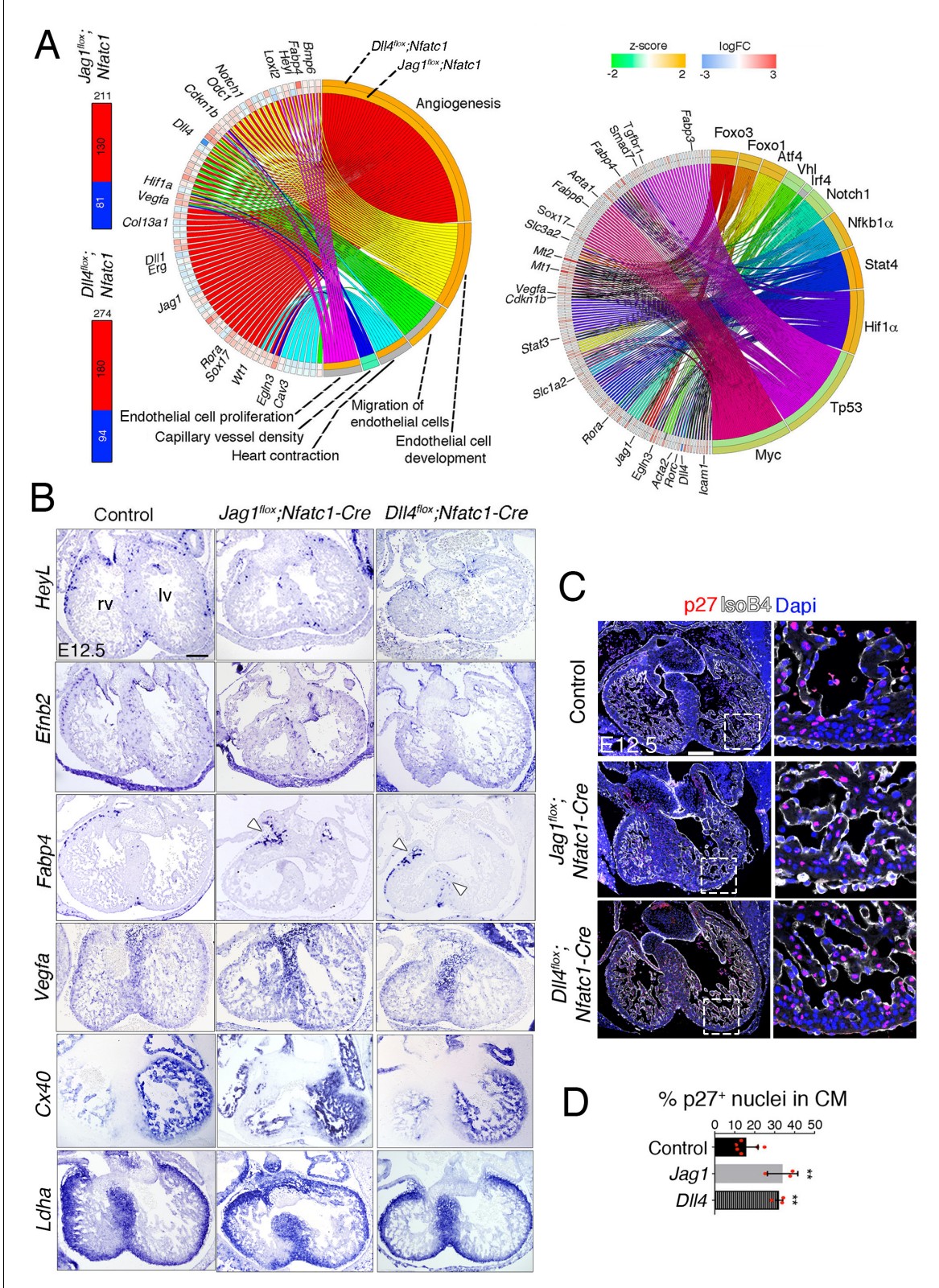

**Figure 2.** Transcriptome profiling of endocardial *Jag1* and *Dll4* mutant hearts. (**A**) *Left*, Total number of differentially expressed genes identified by RNA-seq (Benjamini-Hochberg (**B–H**) adjusted p<0.05) in the indicated genotypes. Numbers indicate upregulated genes (red) and downregulated genes (blue). *Center*, circular plot of representative differentially expressed genes, presenting a detailed view of the relationships between expression changes (left semicircle perimeter) and IPA functions belonging to the Cardiovascular System Development and Function category (right semicircle
*Figure 2 continued on next page*

*Figure 2 continued*

perimeter). For both circular plots, in the left semicircle perimeter, the inner ring represents *Jag1^flox^;Nfatc1-Cre* data and the outer ring *Dll4^flox^;Nfatc1-Cre* data. Activation z-score scale: green, repression; orange, activation; white, unchanged. LogFC scale: red, upregulated; blue, downregulated; white, unchanged. Right, circular plot showing representative differentially expressed genes depending of selected upstream regulators. Details in Table supplement 2 and Table supplement 3. (B) In situ hybridization (ISH) of *HeyL, Efnb2, Fabp4, Vegfa, Cx40* and *Ldha* on E12.5 control, *Jag1^flox^;Nfatc1-Cre,* and *Dll4^flox^;Nfatc1-Cre* heart sections. Arrowheads indicate *Fabp4* expression in capillary vessels. (C) Immunohistochemistry of p27 (red) and IsoB4 (white) on E12.5 control, *Jag1^flox^;Nfatc1-Cre,* and *Dll4^flox^;Nfatc1-Cre* mutant heart sections. Dapi counterstain (blue). Microscope: Nikon A1-R. Software: NIS Elements AR 4.30.02. Build 1053 LO, 64 bits. Objectives: Plan Apo VC 20x/0,75 DIC N2 dry; Plan Fluor 40x/1,3 Oil DIC H N2 Oil. Quantified data for p27-positive nuclei as a % of total CM nuclei. Data are mean ± s.d. (n = 3 sections from 6 control embryos and n = 3 sections from 3 mutant embryos). **p<0.01 by one-way ANOVA with Tukey's multiple comparison tests). Abbreviations: lv, left ventricle; rv, right ventricle. Scale bars, 100 μm.

The online version of this article includes the following figure supplement(s) for figure 2:

**Figure supplement 1.** No evidence of hypoxia in *Jag1^flox^;Nfatc1^Cre^* and *Dll4^flox^;Nfatc1^Cre^* hearts at E12.5.

crossed *Jag1^flox^* and *Dll4^flox^* mice with the vascular endothelium-specific *Pdgfb-iCre^ERT2^* transgenic mice (*Wang et al., 2010*) to obtain the corresponding tamoxifen-inducible lines. Tamoxifen-induced *Jag1* deletion at E12.5 resulted in 68% reduction in *Jag1* expression (*Figure 3—figure supplement 1A,C—Source data 1*, sheet 5) and a complete absence of arteries at E15.5, whereas the veins appeared unaffected (*Figure 3A,C—Source data 1*, sheet 6). We measured NOTCH pathway activity by carrying out a N1ICD staining that showed a 50% increase in endothelial N1ICD (*Figure 3B, C—Source data 1*, sheet 6), consistent with Jag1 acting as an inhibitory Notch ligand. Next, we examined perivascular coverage of the *Jag1^flox^;Pdgfb-iCre^ERT2^* endothelial coronary tree. We used α-smooth muscle actin (αSMA) and Notch3 to measure the extent of coronary vessel smooth muscle cell differentiation and pericyte coverage, respectively (*Volz et al., 2015*). We found that the proportion of αSMA- and Notch3-positive cells was significantly reduced in E15.5 *Jag1^flox^;Pdgfb-iCre^ERT2^* coronary arteries (*Figure 3B,C—Source data 1*, sheet 6) reflecting the lack of differentiation of perivascular cells. Although αSMA is a commonly used marker of vascular smooth muscle cells, it is expressed more broadly in mesenchymal cells and cardiomyocytes throughout the embryonic heart prior to E16.5. To evaluate smooth muscle differentiation more specifically in E15.5 hearts, we used SM22a (*Figure 3—figure supplement 2A,B*). After co-staining with Notch3, we found that the proportion of SM22a and Notch3-positive cells in the *Jag1^flox^;Pdgfb-iCre^ERT2^* mutants was significantly reduced (*Figure 3—figure supplement 2A,B,E—Source data 1*, sheet 7), confirming that pericytes fail to properly differentiate into smooth muscle.

To examine Dll4 function in coronary artery formation, we crossed *Dll4^fllox^* mice with *Pdgfb-iCre^ERT2^* to obtain *Dll4^flox^;Pdgfb-iCre^ERT2^* mice. Tamoxifen-induced *Dll4* deletion from E12.5 or E13.5, resulted in embryonic lethality at E15.5, confirming that embryo survival is highly sensitive to reduction in endothelial *Dll4* expression. However, induction at E14.5 resulted in a 82% reduction in *Dll4* expression (*Figure 3—figure supplement 1B,C—Source data 1*, sheet 5) and a complete absence of arteries at E15.5, whereas veins were unaffected (*Figure 3D,F—Source data 1*, sheet 6). Endothelial N1ICD staining was decreased by 60% compared with controls (*Figure 3E,F—Source data 1*, sheet 6), consistent with Dll4 activating Notch1 during angiogenesis. Furthermore, *Dll4^flox^;Pdgfb-iCre^ERT2^* mutants had deficient perivascular cell coverage as indicated by decreased αSMA- and Notch3-positive cells (*Figure 3E,F—Source data 1*, sheet 6). This was confirmed by co-immunostaining with SM22a and Notch3 demonstrating near complete absence of smooth muscle differentiation in *Dll4^flox^;Pdgfb-iCre^ERT2^* mutants (*Figure 3—figure supplement 2C,D,E—Source data 1*, sheet 7).

## Endocardium/endothelial Notch ligand inactivation impairs coronary artery formation and ventricular growth

To confirm endothelial Jag1 requirement for coronary arterial formation, we crossed *Jag1^flox^* mice with *Cdh5-Cre^ERT2^* driver line (*Wang et al., 2010*) to obtain homozygous *Jag1^flox^;Cdh5^CreERT2^* mice. Tamoxifen-induced *Jag1* inactivation from E9.5 onwards (*Figure 3—figure supplement 3A*) resulted in reduced coronary artery coverage and marginally increased vein coverage (*Figure 3—figure supplement 3A,C—Source data 1*, sheet 8). N1ICD staining in the endothelium was increased (*Figure 3—figure supplement 3B,C—Source data 1*, sheet 8), reflecting Jag1 inhibitory Notch function during angiogenesis. Furthermore, *Jag1^flox^;Cdh5^CreERT2^* mutant heart coronaries had decreased

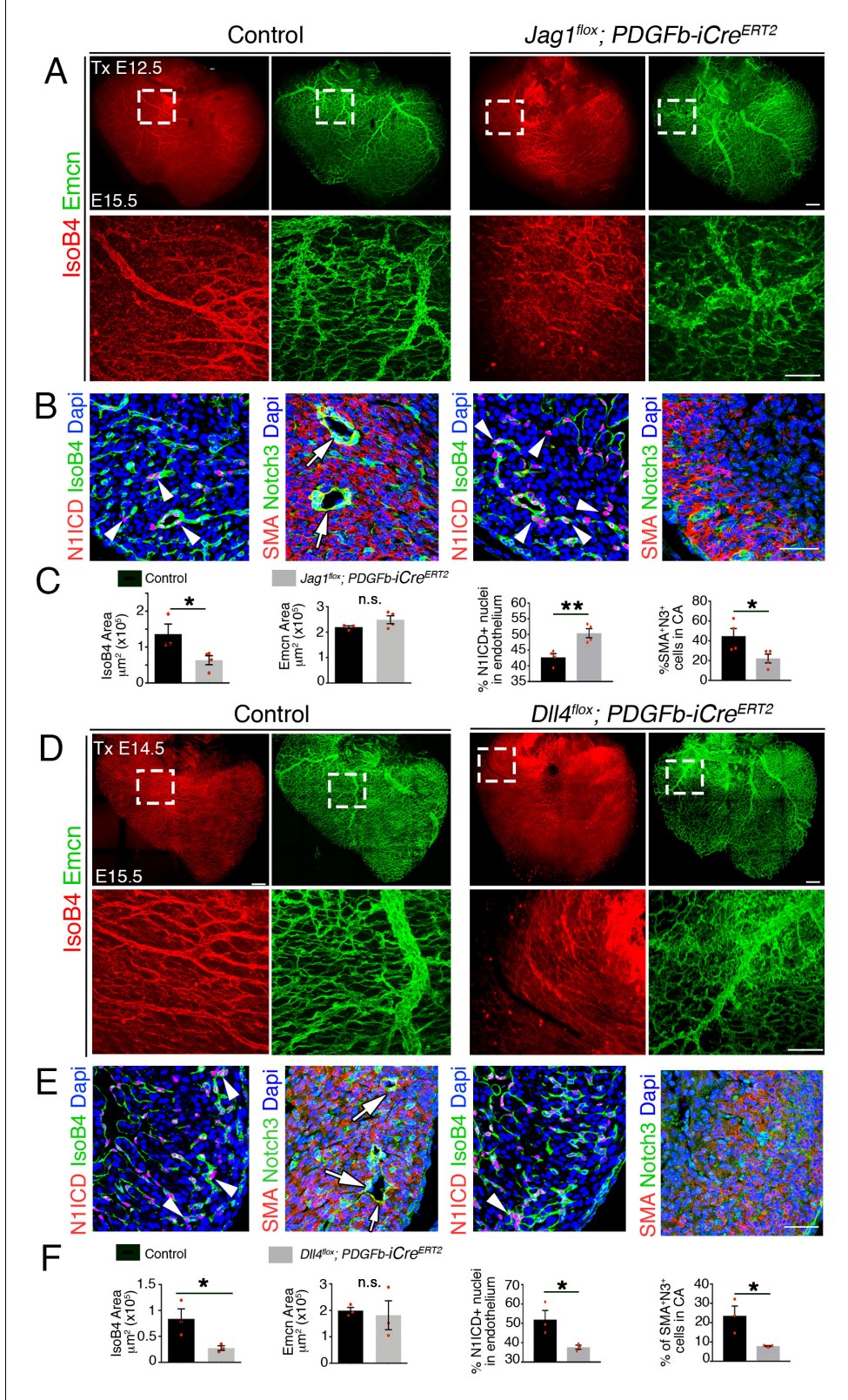

**Figure 3.** Late endothelial Jag1 or Dll4 inactivation disrupts coronary plexus remodeling. (**A**) Dorsal view of whole-mount immunochemistry for IsoB4 (red), labelling arteries and Emcn (green), labeling veins and capillaries, in E15.5 control and *Jag1^flox^;Pdgfb-iCre^ERT2^* mutant hearts, tamoxifen (Tx)-induced at E12.5. Scale bars, 100 µm. (**B**) E15.5 control and *Jag1^flox^;Pdgfb-iCre^ERT2^* mutant heart sections. Left, Immunostaining for N1ICD (red) and IsoB4 (green). *Figure 3 continued on next page*

*Figure 3 continued*

Right, α-smooth-muscle actin (SMA, red) and Notch3 (green). Dapi counterstain (blue). Arrowheads point to N1ICD-positive nuclei. Arrows point to αSMA-Notch3 co-immunostaining. Scale bars, 100 μm. (**C**) Quantified data from E15.5 control and *Jag1*$^{flox}$*;Pdgfb-iCre*$^{ERT2}$ hearts: area of coverage by coronary arteries (IsoB4-positive vessels) and veins (Emcn-positive vessels); Data are mean ± s.d. (n = 3 control embryos and 4 mutant embryos. N1ICD-positive nuclei as a percentage of total endothelial nuclei; and SMA-Notch3 co-immunostaining in coronary arteries. Data are mean ± s.d. (n = 3 sections from 4 control embryos and from 4 mutant embryos. (**D**) Whole-mount dorsal view of immunohistochemistry for Emcn (green) and IsoB4 (red) in E15.5 control and *Dll4*$^{flox}$*;Pdgfb-iCre*$^{ERT2}$ mutant hearts. Tx-induced at E14.5. (**E**) Left, immunohistochemistry for N1ICD (red) and IsoB4 (green); right, immunohistochemistry for SMA (red) and Notch3 (green) on control E15.5 WT and *Dll4*$^{flox}$*;Pdgfb-iCre*$^{ERT2}$ mutant heart sections. Dapi counterstain (blue). Arrowheads indicate N1ICD-positive nuclei. Arrows point to SMA-Notch3 co-immunostaining. Scale bars, 100 μm. Microscopes: Nikon A1-R, Leica SP5. Softwares: NIS Elements AR 4.30.02. Build 1053 LO, 64 bits (Nikon); LAS-AF 2.7.3. build 9723 (Leica). Objectives: Plan Apo VC 20x/0.75 DIC N2 dry; Plan Fluor 40x/1.3 Oil DIC H N2 Oil (Nikon); HCX PL APO CS 10 × 0.4 dry. HCX PL APO lambda blue 20 × 0.7 multi-immersion (Leica). (**F**) Quantified data from E15.5 control and *Dll4*$^{flox}$*;Pdgfb-iCre*$^{ERT2}$ hearts: area covered by coronary arteries (IsoB4-positive vessels); area covered by veins (Emcn-positive vessels); percentage of N1ICD-positive nuclei in endothelium as a percentage (%) of total nuclei; and SMA-Notch3 co-immunostaining in coronary arteries. Data are mean ± s.d. (n = 3 sections from 3 control embryos and from three mutant embryos). *p<0.05, **p<0.01, by Student's t-test; n.s., not significant. Scale bars, 100 μm.

The online version of this article includes the following figure supplement(s) for figure 3:

**Figure supplement 1.** Efficient tamoxifen-induced endothelial deletion of *Jag1* and *Dll4* using *Pdgfb-*$^{iCreERT2}$ driver line.

**Figure supplement 2.** Induced endothelial *Jag1* or *Dll4* deletion disrupts coronary smooth muscle cell differentiation.

**Figure supplement 3.** Late endothelial *Jag1* or *Dll4* inactivation disrupts coronary plexus remodeling.

**Figure supplement 4.** Induced endothelial *Jag1* or *Dll4* deletion disrupts myocardial growth.

---

perivascular cell coverage, as indicated by the reduced proportion of αSMA- and Notch3-positive cells (***Figure 3—figure supplement 3B,C—Source data 1***, sheet 8).

To confirm Dll4 requirement for coronary arterial formation and circumvent the lethality resulting from E12.5 tamoxifen administration in *Dll4*$^{flox}$*;Pdgfb-iCre*$^{ERT2}$ mice, we crossed *Dll4*$^{flox}$ mice with *Cdh5-Cre*$^{ERT2}$ driver line (***Wang et al., 2010***) to obtain homozygous *Dll4*$^{flox}$*;Cdh5-Cre*$^{ERT2}$ mice (***Figure 3—figure supplement 3D***). Tamoxifen induction at E12.5 resulted in the reduction of coronary arterial coverage, unchanged vein coverage but reduced vein caliber (***Figure 3—figure supplement 3D,F—Source data 1***, sheet 8). N1ICD staining in the endothelium was decreased by 60% compared with controls (***Figure 3—figure supplement 3E,F—Source data 1***, sheet 8), likely due to reduced EC contribution to the arteries. Furthermore, *Dll4*$^{flox}$*;Cdh5-Cre*$^{ERT2}$ mutant heart coronaries had decreased perivascular cell coverage as indicated by the reduced proportion of αSMA- and Notch3-positive cells (***Figure 3—figure supplement 3E,F—Source data 1***, sheet 8). The impact of impaired coronary vessel formation in endothelial *Jag1-* or *Dll4-* mutants was also indicated by their diminished ventricular wall thickness (***Figure 3—figure supplement 4A—Source data 1***, sheet 9).

To examine the effect of endothelial *Dll4* deletion on cardiac gene expression, we performed RNA-seq on E15.5 *Dll4*$^{flox}$*;Cdh5-Cre*$^{ERT2}$ mutant ventricles. Therefore of 163 DEGs, 138 were upregulated, while 25 were downregulated (***Figure 3—figure supplement 4B*** and ***Supplementary file 2***, sheet 3). As expected, *Dll4* was downregulated, while *Vegfa* was upregulated, suggesting cardiac hypoxia (***Figure 3—figure supplement 4B—Supplementary file 2***, sheet 3). *Cdkn1a*/p21 was upregulated, indicating impaired cell proliferation (***Figure 3—figure supplement 4B—Supplementary file 2***, sheet 3). IPA identified strong upregulation of endothelial cell functions (***Figure 3—figure supplement 4B*** and ***Supplementary file 3***), whereas analysis of upstream regulators identified an enrichment for transcriptional activators associated with innate inflammatory responses (Irfs and Stats), suggesting defective vascular integrity. Other upstream regulators were associated with cardiovascular disease, including hypertrophy (Nfatc2) and oxidative stress responses (Nfe212; ***Figure 3—figure supplement 4B*** right and ***Supplementary file 3***). Moreover, ISH showed reduced endothelial expression of *Efnb2* and *Dll4* in smaller caliber vessels (***Figure 3—figure supplement 4C***), suggesting a loss of arterial identity, whereas *Fapb4* was upregulated (***Figure 3—figure supplement 4C***) consistent with increased coronary vessel density of the un-remodeled coronary plexus.

Thus, endothelial Jag1 or Dll4 signaling promotes remodeling and maturation of the coronary vascular tree, and in the absence of adequate vascular remodeling, cardiac growth is impaired.

## Forced endothelial *Dll4* expression disrupts coronary vascular remodeling

To further characterize the role of Notch in coronary development, we generated a transgenic line (*Dll4$^{GOF}$*) bearing a *Rosa26-CAG-floxNeoSTOPflox-Dll4-6xMycTag* expression cassette (see Materials and methods). *Tie2-Cre*-mediated removal of the *floxed NeoSTOP* sequences resulted in a mild 1.2-fold endothelial *Dll4* overexpression that permitted survival of transgenic embryos bearing a single copy of the *Dll4$^{GOF}$* allele (not shown). At E14.5, forced *Dll4* expression lead to marginally increased, sub-epicardial vessel (Emcn-positive) coverage (*Figure 4A,F—Source data 1*, sheet 10) and vascular malformations similar to those found in *Dll4$^{flox}$;Pdgfb-iCre$^{ERT2}$* and *Dll4$^{flox}$;Cdh5-Cre$^{ERT2}$* mutants (*Figure 1—figure supplement 3E*). However, by E16.5 IsoB4-positive (arteries) intramyocardial vessels were substantially decreased (*Figure 4B,F—Source data 1*, sheet 10). Accordingly, *Dll4* and *Efnb2* expression was restricted to smaller caliber vessels in *Dll4$^{GOF}$* transgenics (*Figure 4C*). In contrast, superficial Emcn-positive vessels (veins) were more numerous and dense (*Figure 4B,F—Source data 1*, sheet 10). N1ICD expression was increased and extended to the prospective veins (*Figure 4D,G—Source data 1*, sheet 10), consistent with Notch1 gain-of-function in endothelium. Arterial smooth muscle cell coverage was also substantially reduced (*Figure 4E,G—Source data 1*, sheet 10), suggesting defective pericyte and/or smooth muscle differentiation. These coronary vascular defects were associated with below-normal cardiomyocyte proliferation and reduced myocardial thickness (not shown). Therefore, endothelial *Dll4* overexpression blocks coronary artery formation, vessel remodeling and maturation, and impairs cardiac growth. These results were surprising, as our loss-of-function data support a pro-arteriogenic role for Dll4.

To support our *Dll4$^{GOF}$* results, we used a transgenic line conditionally overexpressing Mfng (*Mfng$^{GOF}$*) (*D'Amato et al., 2016*) to test whether increased Mfng activity favored Dll4-mediated signaling, and thus impaired arteriogenesis. At E11.5, *Mfng$^{GOF}$;Tie2-Cre* embryos displayed reduced SV sprouting, increased Notch1 signaling and myocardial wall thinning (*Figure 4—figure supplement 1A,B*). However, by E14.5 *Mfng$^{GOF}$;Tie2-Cre* embryos displayed an extensive vascular network (*Figure 4—figure supplement 1C*), although the arteries appeared atrophied (*Figure 4—figure supplement 1C*) and prospective veins appeared more numerous and intricately branched (*Figure 4—figure supplement 1C*). *Mfng$^{GOF}$;Tie2-Cre* embryos displayed numerous capillary malformations at E14.5 (*Figure 1—figure supplement 3F*), suggesting defective arterial-venous differentiation. The decrease in arterial coverage and increase in venous coverage in *Mfng$^{GOF}$;Tie2-Cre* embryos became more obvious by E16.5 (*Figure 4—figure supplement 1C,G—Source data 1*, sheet 11). N1ICD expression was markedly increased (*Figure 4—figure supplement 1D,G—Source data 1*, sheet 11), as expected, while the expression of the Notch targets *HeyL* and *Efnb2* was restricted to smaller caliber vessels (*Figure 4—figure supplement 1E*). *Vegfa* appeared upregulated throughout (*Figure 4—figure supplement 1E*), suggesting that *Mfng$^{GOF}$;Tie2-Cre* hearts are hypoxic. Pericyte coverage and smooth muscle differentiation, as shown by αSMA and Notch3 co-immunostaining, were significantly decreased at E16.5 (*Figure 4—figure supplement 1F,G—Source data 1*, sheet 11), suggesting defective pericyte recruitment and/or differentiation. Ink injection confirmed that *Mfng$^{GOF}$;Tie2-Cre* coronaries were 'leaky' (*Figure 4—figure supplement 1H*), and therefore functionally deficient.

## EphrinB2 is required for coronary arteriogenesis and vessel branching

*Efnb2* is necessary for arterial-venous differentiation and vascular maturation (*Kania and Klein, 2016*) and is a Notch target during ventricular chamber development (*Grego-Bessa et al., 2007*). We find *Efnb2* expression in ventricular endocardium at E10.5 (*Figure 5—figure supplement 1A*) but not in SV endothelium (not shown). At E13.5, *Efnb2* was expressed in emerging arteries and prospective veins (*Figure 5—figure supplement 1A*), and at E16.5 was confined to arteries (*Figure 5—figure supplement 1A*). Therefore, *Efnb2* is expressed dynamically in a pattern similar to *Dll4* and *Mfng*.

To examine EphrinB2 function in coronary angiogenesis, we crossed mice bearing a conditional *Efnb2$^{flox}$* allele with the *Nfatc1-Cre* line. Whole-mount IsoB4 immunostaining revealed reduced

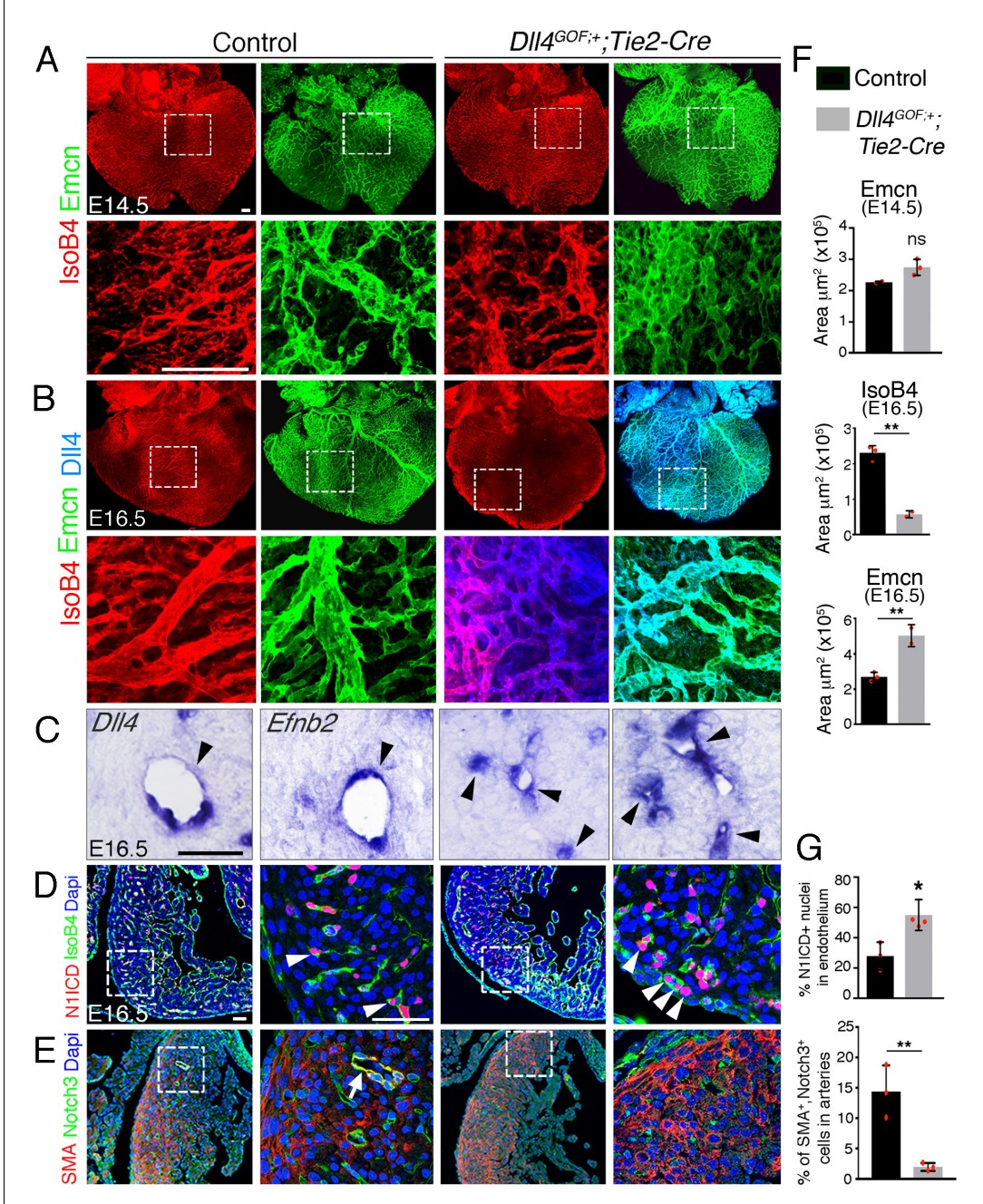

**Figure 4.** Forced Dll4 expression in endothelium disrupts coronary arteriovenous differentiation and remodeling. (**A**) Dorsal view of whole-mount immunochemistry for IsoB4 (red), labeling arteries, and Emcn (green), labeling veins and capillaries, in E14.5 control and *Dll4^{GOF}*;*Tie2-Cre* heart. Scale bar, 100 µm. (**B**) Dorsal view of whole-mount immunochemistry for IsoB4 (red) and Emcn (green) in E16.5 control and *Dll4^{GOF}*;*Tie2-Cre* mutant heart. Scale bar, 100 µm. (**C**) ISH of *Dll4* and *Efnb2* on E16.5 control and *Dll4^{GOF}*;*Tie2-Cre* heart sections. Arrowheads indicate coronary arteries. Scale bar, 50 µm. (**D**) Immunohistochemistry for N1ICD (red) and IsoB4 (green) on E16.5 control and *Dll4^{GOF}*;*Tie2-Cre* heart sections. Dapi counterstain (blue). Arrowheads indicate N1ICD-stained nuclei. Scale bar, 50 µm. (**E**) Immunohistochemistry for SMA (red) and Notch3 (green) on E16.5 control and *Dll4^{GOF}*; *Tie2-Cre* heart sections. Dapi counterstain (blue). The arrow points to a coronary vessel stained by SMA and Notch3. Scale bar, 50 µm. Microscope: Nikon A1-R. Software: NIS Elements AR 4.30.02. Build 1053 LO, 64 bits. Objectives: Plan Apo VC 20x/0,75 DIC N2 dry; Plan Fluor 40x/1,3 Oil DIC H N2 Oil. (**F**) Quantified data for control and *Dll4^{GOF}*;*Tie2-Cre* hearts: E14.5, area covered by veins (Emcn-positive vessels). Data are mean ± s.d, (n = 2 control embryos and n = 3 mutant embryos); E16.5, area covered by coronary arteries (IsoB4-positive vessels), and area covered by veins (Emcn-positive vessels). Data are mean ± s.d, (n = 3 control embryos and n = 2 mutant embryos). (**G**) Quantified data of the percentage of N1ICD-positive nuclei in E16.5 endothelium as a percentage (%) of total nuclei; and SMA-Notch3 co-immunostaining in coronary arteries. Data are mean ± s.d, (n = 3 sections from 3control embryos and n = 4 sections from 3 mutant embryos) *p<0.05, **p<0.01, by Student's *t*-test; ns. not significant.
*Figure 4 continued on next page*

Figure 4 continued

The online version of this article includes the following figure supplement(s) for figure 4:

**Figure supplement 1.** Forced *Mfng* expression disrupts arterio-venous differentiation and arterial endothelial integrity.

artery coverage in E15.5 *Efnb2^flox^;Nfatc1-Cre* hearts (*Figure 5A,F—Source data 1*, sheet 12), while vein coverage (Emcn-positive) was unchanged (*Figure 5A,F—Source data 1*, sheet 12). Histological examination showed that the compact myocardium in the ventricles was 30% thinner at E16.5 (*Figure 5—figure supplement 1B,C—Source data 1*, sheet 13), which could be attributed to the reduced proliferation observed at E14.5 (*Figure 5—figure supplement 1C,D—Source data 1*, sheet 13). The presence of smaller caliber arteries coincided with higher Notch1 activity (*Figure 5B,F—Source data 1*, sheet 12) and increased *Hey1* expression (*Figure 5C*), suggesting negative feedback between EphrinB2 and Notch signaling. Although the patterning of prospective veins was unchanged, we found occasional malformations (*Figure 1—figure supplement 3G,H*) and transmural communications or shunts between the endocardium and epicardium, which we interpret to be arteriovenous fistulae (*Figure 5—figure supplement 1E, F*).

Accordingly, the venous marker EphB4 was expressed ectopically in a subset of intramyocardial vessels at E16.5 (*Figure 5—figure supplement 1E*), indicative of abnormal venous identity. *Vegfa* was upregulated throughout the heart (*Figure 5—figure supplement 1F*), suggesting that *Efnb2^flox^;Nfatc1-Cre* hearts are hypoxic. Arterial smooth muscle coverage was reduced substantially (*Figure 5E,F—Source data 1*, sheet 12), suggesting defective vessel maturation. Dye perfusion in *Efnb2^flox^;Nfatc1-Cre* hearts revealed absence of left anterior descending coronary artery at E16.5 (*Figure 5—figure supplement 1G*) and ink perfusion revealed vascular hemorrhaging in the most distal section of the coronary arterial tree at E18.5 (*Figure 5—figure supplement 1H*), consistent with vessel leakiness, while aortic connections were normal (*Figure 5—figure supplement 1H*). Thus, endocardial EphrinB2 is required for coronary arterial remodeling, vessel structural integrity, and cardiac growth.

## EphrinB2 mediates Jag1 and Dll4 signaling in arterial branching morphogenesis

To model coronary angiogenesis ex vivo, we developed a ventricular explant assay (*Figure 6A*; (*Zhang and Zhou, 2013*). Compared with controls, *Jag1^flox^;Nfatc1-Cre* explants had a lower angiogenic potential, manifested as smaller caliber vessels, longer distances between branching points and a lower number of transmural endothelial branches (*Figure 6B,C,K—Source data 1*, sheet 14). Conversely, *Dll4^flox^;Nfatc1-Cre* or *Notch1^flox^;Nfatc1-Cre* explants yielded more densely interconnected vascular networks of larger vessel caliber, longer distances between vessel branching points, and more transmural endothelial branches (*Figure 6B,D,E,K—Source data 1*, sheet 14). *Notch1^flox^;Nfatc1-Cre* explants displayed a phenotype similar to *Dll4^flox^;Nfatc1-Cre* explants, although the differences in vessel caliber and endothelial branch number did not reach significance (*Figure 6B,E,K—Source data 1*, sheet 14). *Mfng^GOF^;Tie2-Cre* and *Dll4^GOF^;Tie2-Cre* explants showed a broadly similar phenotype, with substantial lengthening of branching point distance and a below-normal number of endothelial branches (*Figure 6B,I,J,K—Source data 1*, sheet 14). In contrast, *Mfng^GOF^;Tie2-Cre* explants had smaller caliber vessels than *Dll4^GOF^;Tie2-Cre* explants (*Figure 6I,J,K—Source data 1*, sheet 14). *Efnb2^flox^;Nfatc1-Cre* explants formed sparsely interconnected networks with decreased vessel caliber, reduced number of transmural endothelial branches, and increased distances between branching points, similar to those seen in *Jag1^flox^;Nfatc1-Cre* explants (*Figure 6B,F,K—Source data 1*, sheet 14).

Based on the congruence of endocardial/endothelial *Jag1*, *Dll4* and *Efnb2* loss-of-function phenotypes, we tested whether EphrinB2 mediates Notch function in coronary angiogenesis. Thus, delivery of a lentivirus expressing *Efnb2* to *Jag1^flox^;Nfatc1-Cre* ventricular explants normalized the reduced endothelial branch number to the number seen in control explants (*Figure 6B,G,K—Source data 1*, sheet 14). Likewise, *Efnb2* expression restored the elevated number of endothelial branches in *Dll4^flox^;Nfatc1-Cre* explants to control levels (*Figure 6B,H,K—Source data 1*, sheet 14), showing that the branching defect could be normalized in both mutants.

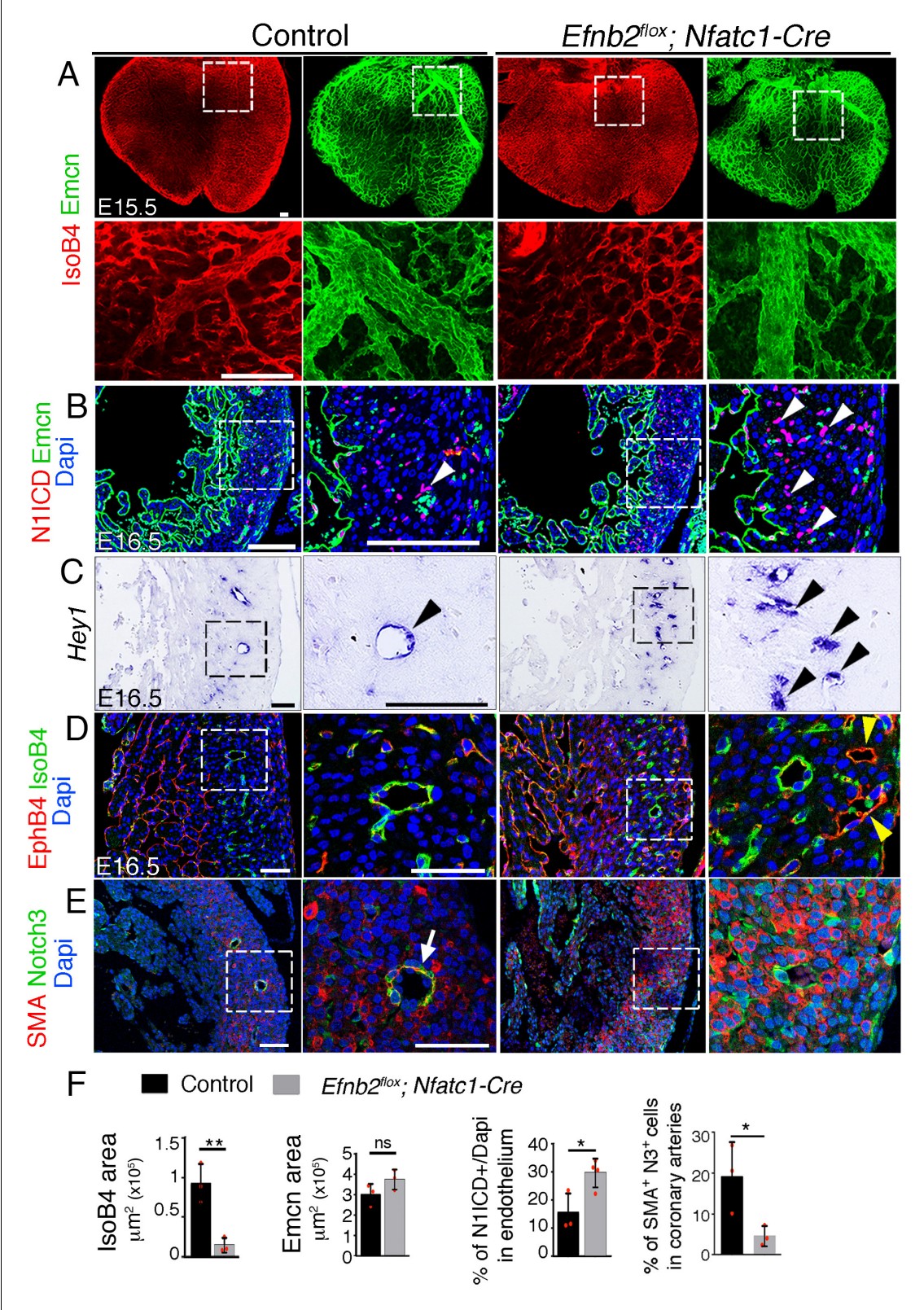

**Figure 5.** Endocardial *Efnb2* inactivation disrupts coronary artery differentiation and remodeling. (**A**) Dorsal view of whole-mount immunochemistry for IsoB4 (red), labelling arteries and Emcn (green), labeling veins and capillaries, in E15.5 control and *Efnb2flox;Nfatc1-Cre* hearts. Scale bars,100 µm. (**B**) Immunohistochemistry of N1ICD (red) and Emcn (green) on E16.5 control and *Efnb2flox;Nfatc1-Cre* heart sections. Dapi-counterstain (blue). Arrowheads indicate N1ICD-stained nuclei. Scale bars,100 µm. (**C**) ISH of *Hey1* on E16.5 control and *Efnb2flox;Nfatc1-Cre* heart sections. Arrowheads indicate *Hey1-*

*Figure 5 continued on next page*

*Figure 5 continued*

expressing coronaries. Scale bars, 100 μm. (**D**) Immunohistochemistry for EphB4 (red) and IsoB4 (green) on E16.5 control and *Efnb2^flox^;Nfatc1-Cre* heart sections. Dapi-counterstain (blue). Yellow arrowheads indicate EphB4-stained vessels. Scale bar, 50 μm. (**E**) Immunohistochemistry for SMA (red) and Notch3 (green) on E16.5 control and *Efnb2^flox^;Nfatc1-Cre* heart sections. Dapi-counterstain (blue). Arrow points to a coronary artery stained by SMA and Notch3. Microscope: Nikon A1-R. Software: NIS Elements AR 4.30.02. Build 1053 LO, 64 bits. Objectives: Plan Apo VC 20x/0,75 DIC N2 dry; Plan Fluor 40x/1,3 Oil DIC H N2 Oil. (**F**) Quantified data for control and E16.5 WT and *Efnb2^flox^;Nfatc1-Cre* hearts: E15.5, area covered by coronary arteries (IsoB4-positive vessels), area covered by veins (Emcn-positive vessels). Data are mean ± s.d, (n = 3 control embryos and n = 3 mutant embryos), and percentage of N1ICD-stained nuclei in endothelium as percentage (%) of total nuclei Data are mean ± s.d. (n = 3 sections from 3 control embryos and n = 3 sections from 4 mutant embryos); E16.5, SMA-Notch3 co-immunostaining in coronary arteries. Data are mean ± s.d. (n = 3 sections from three control embryos and n = 3 sections from 3 mutant embryos). *p<0.05, **p<0.01 by Student's *t*-test; n.s., not significant.

The online version of this article includes the following figure supplement(s) for figure 5:

**Figure supplement 1.** Endocardial *Efnb2* deletion disrupts myocardial proliferation, arterio-venous differentiation, and arterial endothelial integrity.

## EphrinB2 functions downstream of Jag1 and Dll4 in capillary tube formation

The ventricular explant assay assesses the collective behaviors of endocardium and coronary endothelial outgrowth. In order to determine endothelial behavior exclusively, we performed capillary tube formation assays with primary human endothelial cells (HUVEC). shRNA directed against *JAG1*, *DLL4* or *EFNB2* were lentivirally-transduced into HUVEC. This resulted in reduction of mRNA levels of 30% and 60% for *JAG1* and *DLL4*, respectively (*Figure 7A—Source data 1*, sheet 15), and of 50% for *EFNB2* (*Figure 7—figure supplement 1A—Source data 1*, sheet 16). The NOTCH target *HEY1* was unchanged in *shJAG1*-infected cells and 50% decreased in *shDLL4*-infected cells (*Figure 7A—Source data 1*, sheet 15). The activity of *EFNB2* lentivirus was verified by rescuing shRNA-mediated *EFNB2* knockdown (*Figure 7—figure supplement 1B,C—Source data 1*, sheet 16). *EFNB2* knockdown resulted in decreased capillary network complexity (*Figure 7—figure supplement 1B,C—Source data 1*, sheet 16). These parameters were restored to control levels by lentiviral-mediated *EFNB2* overexpression (*Figure 7—figure supplement 1B,C—Source data 1*, sheet 16).

We next tested whether EPHRINB2 acts downstream of JAG1 and DLL4 in capillary tube formation. *JAG1* knockdown inhibited capillary tube formation and decreased network complexity as determined by measuring EC junctions, nodes, segments and pieces that were below control, while the 'total isolated branches length' was above control (*Figure 7B,C—Source data 1*, sheet 15). However, these parameters were restored to control levels in presence of the *EFNB2* transgene (*Figure 7B,C—Source data 1*, sheet 15), suggesting that EPHRINB2 compensates for *JAG1* knockdown in this assay. In contrast, knockdown of *DLL4* increased capillary tube formation and network complexity (junctions, nodes, segments and pieces; *Figure 7B,C—Source data 1*, sheet 15), while measurements were restored and even went beyond control by overexpressing *EFNB2* (*Figure 7B, C—Source data 1*, sheet 15). Thus, *EFNB2* overexpression not only compensates for the absence of DLL4 in this assay, but has an added effect as well.

Taken together, our observations with ventricular explants and HUVEC indicate that JAG1 and EPHRINB2 promote coronary vessel sprouting and branching, whereas DLL4, MFNG and NOTCH1 inhibit these processes. Moreover, EPHRINB2 mediates signaling from both JAG1 and DLL4 to regulate endothelial branching during coronary angiogenesis.

## Discussion

The origin of the coronary endothelium from a venous source has prompted the suggestion that arteries need to be 'reprogrammed' from veins (*Red-Horse et al., 2010*). We found Jag1, Dll4, Mfng, and N1ICD expression in SV endothelium and sub-epicardial capillaries, with Dll4 and Mfng expression maintained subsequently in prospective veins. These Notch pathway elements are co-expressed with the venous marker endomucin, which is also expressed in the endocardium and SV (*Cavallero et al., 2015*). This implies that nascent subepicardial vessels have a mixed arterial/venous identity. During embryonic angiogenesis, Notch is necessary for artery-vein specification—the reversible commitment of ECs to arterial or venous fate—before the onset of blood flow (*Swift and Weinstein, 2009*). Our finding of SV endothelial cell heterogeneity in relation to arterial-venous identity is consistent with the notion that early vascular beds are phenotypically plastic during embryonic

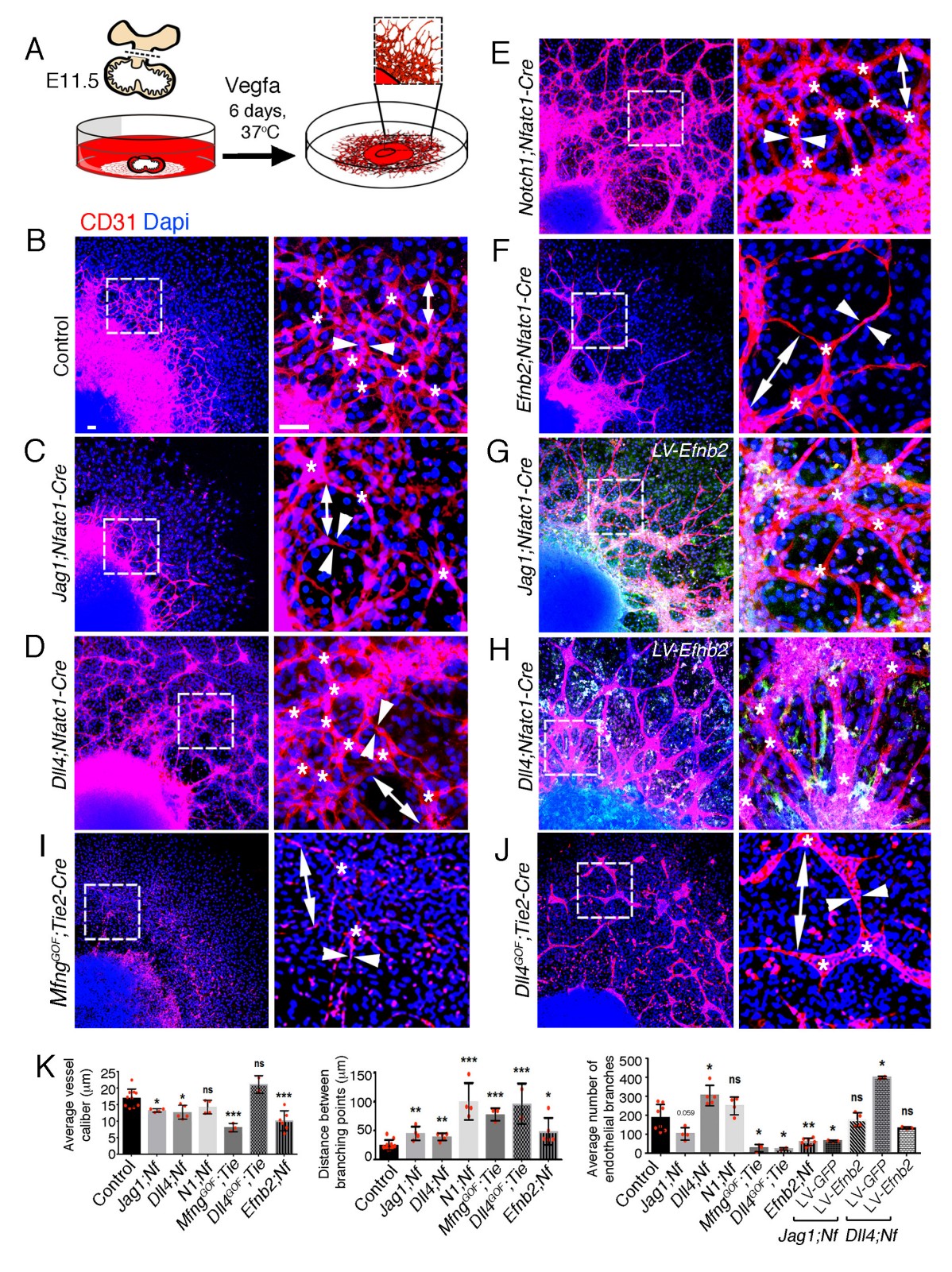

**Figure 6.** EPHRINB2 rescues disrupted arterial branching in ventricular explants from *Jag1* and *Dll4* mutant hearts. (**A**) Ventricular explant assay procedure. (**B–J**) Representative images of E11.5 cultured ventricular explants from the following embryos: (**B**) control (n = 8–10), (**C**) *Jag1flox;Nfatc1-Cre* (n = 4), (**D**) *Dll4flox;Nfatc1-Cre* (n = 5), (**E**) *Notch1flox;Nfatc1-Cre* (n = 4), (**F**) *MfngGOF;Tie2-Cre* (n = 3), (**G**) *Dll4GOF;Tie2-Cre* (n = 3), (**H**) *Efnb2flox;Nfatc1-Cre* (n = 5–7), (**I**) *Dll4flox;Nfatc1-Cre* infected with *Efnb2*-overexpressing lentivirus (n = 5–7). Microscope: Nikon A1-R. Software: NIS Elements AR 4.30.02. *Figure 6 continued on next page*

*Figure 6 continued*

Build 1053 LO, 64 bits. Objectives: Plan Apo VC 20x/0,75 DIC N2 dry; Plan Fluor 40x/1,3 Oil DIC H N2 Oil. (**J**) Quantification of vessel caliber (arrowheads), branching point distance (double ended arrowhead), and mean endothelial branch (asterisk) number. Data are means ± s.d. ***p<0.001; **p<0.01; *p<0.05, by Student's t-test; Benjamini-Hochberg adjusted p-value. n.s not significant. Scale bars, 100 µm.

development (*Moyon et al., 2001*; *Chong et al., 2011*; *Fish and Wythe, 2015*). Our results are consistent with recently published data showing that pre-arterial cells expressing Notch pathway elements are present in coronary endothelium (*Su et al., 2018*), prior to blood flow onset.

We show that Notch ligands Jag1 and Dll4 are required for SV sprouting angiogenesis, further supporting the notion that coronary arteries are specified by Notch prior to blood flow (*Su et al., 2018*) (*Figure 8A*). Dll4 is the key Notch ligand activator regulating embryonic (*Duarte, 2004*; *Gale et al., 2004*; *Krebs et al., 2004*; *Benedito et al., 2008*) and postnatal retinal angiogenesis (*Hellström et al., 2007*; *Lobov et al., 2007*), whereas Jag1 antagonizes Dll4-Notch1 signaling (*Benedito et al., 2009*) and acts downstream of Dll4-Notch1 to promote smooth muscle differentiation (*Pedrosa et al., 2015*). Jag1 or Dll4 inactivation in the SV results in arrested and excessive angiogenesis, respectively, resembling the situation in the retina (*Benedito et al., 2009*) (*Figure 8B*). Moreover capillary 'entanglements' suggestive of vascular malformations were present in the early plexus of *Jag1^flox;Nfatc1-Cre* and *Dll4^flox;Nfatc1-Cre* mutants. Vascular malformations in Notch mutants have been attributed to unresolved intermingling of arteries and veins during differentiation, or to failed maintenance of arterial and venous identities in the vascular bed (*Gale et al., 2004*; *Krebs et al., 2004*). The presence of these malformations in *Jag1^flox;Nfatc1-Cre* and *Dll4^flox;Nfatc1-Cre* mutants could therefore indicate that endothelial progenitors begin to differentiate into arteries and veins very soon after exiting the SV.

Consistent with the known roles of the Notch ligands in angiogenesis, the hierarchical organization of the coronary vascular tree was profoundly altered in the late-induced *Jag1* and *Dll4* mutants, implying that both Notch ligands are required for high-order complexity of the coronary vessels (*Figure 8C*). Jag1 or Dll4 inactivation during coronary arterial remodeling results in smaller diameter arteries, consistent with Notch promoting vascular remodeling and arterial fate commitment, as described in other developmental settings (*Figure 8D*; (*Duarte, 2004*; *Gale et al., 2004*; *Krebs et al., 2004*). However, gain of Notch function also led to smaller caliber arteries (*Figure 8D*), when the opposite outcome was anticipated (*Uyttendaele et al., 2001*; *Trindade et al., 2008*; *Krebs et al., 2010*). This result implies that non-physiological variations of Notch activity lead to the arrest of coronary artery differentiation. *Dll4* mutants also showed coronary vessel hemorrhaging reminiscent of that found in vascular disorders (*Park-Windhol and D'Amore, 2016*) or tumors (*Goel et al., 2011*) and indicative of disrupted endothelial integrity.

We reasoned that the coronary maturation defects observed in *Jag1* or *Dll4* mutants are due, at least in part, to the loss of EphrinB2 function given that *Efnb2* is a direct Dll4-Notch signaling target in ECs (*Iso et al., 2006*) and endocardium during ventricular development (*Grego-Bessa et al., 2007*). Moreover, EphrinB2 and VEGF are functionally linked during angio- and lymphangiogenesis; EphrinB2 is a direct activator of VEGFR2 and VEGFR3, and therefore cooperates in the mechanism leading to tip cell extension and vessel sprouting (*Sawamiphak et al., 2010*; *Wang et al., 2010*). Thus, *Efnb2* inactivation leads to coronary artery remodeling defects, similar to those resulting from *Jag1* or *Dll4* inactivation, suggesting that EphrinB2 functions downstream of Notch to promote coronary arterial remodeling (*Figure 8D*). This notion is supported by the angiogenic ventricular explant and EC capillary tube assays, in which opposite effects of *Jag1* or *Dll4* deficiency on vessel branching are rescued by transduction of an *EFNB2*-expressing lentivirus, identifying EphrinB2 as a Notch effector during coronary artery development.

A common feature among the endocardial or endothelial Notch loss- and gain-of-function models analyzed in our study is the thin ventricular wall (*Figure 8B,D*). Of interest is that myocardial inactivation of *Jag1*, or combined inactivation of *Jag1* and *Jag2*, or *Mib1*, leads to thinner ventricular walls, accompanied by reduced cardiomyocyte proliferation, disrupted ventricular chamber patterning, and cardiomyopathy (*Luxán et al., 2013*; *D'Amato et al., 2016*). In contrast, chamber patterning is maintained in endocardial *Jag1*, *Dll4*, and *Efnb2* mutants and endothelial *Dll4* and *Jag1* mutants. Our results suggest that disturbed ventricular wall growth in the earlier E11.5-E12.5 *Jag1^flox;Nfatc1-Cre* is caused by altered endocardial-myocardial signaling, as suggested for *Dll4^flox;Nfatc1-Cre*

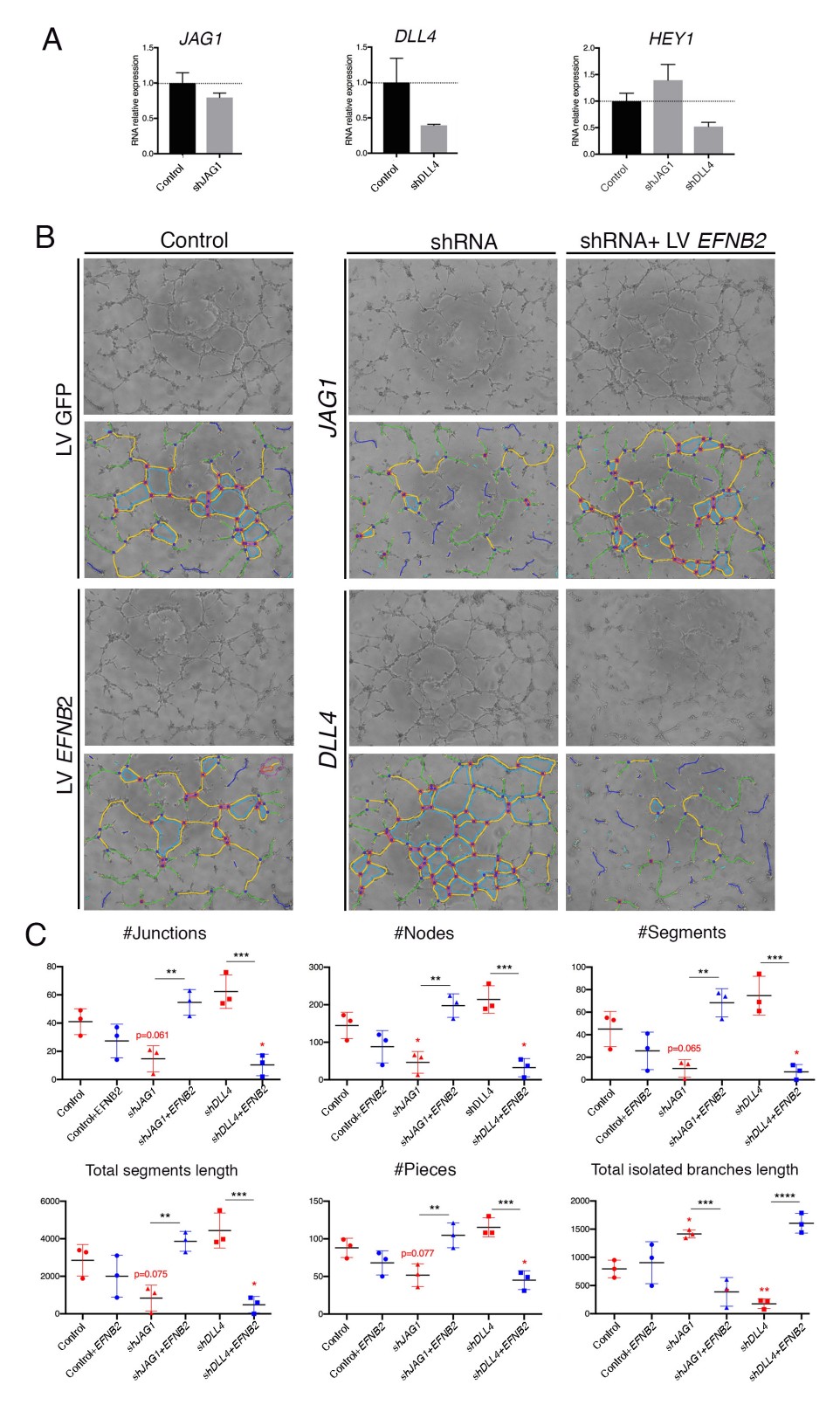

**Figure 7.** EPHRINB2 rescues defective capillary network formation resulting from shRNA-mediated silencing of JAG1 and DLL4 in HUVEC. (**A**) qRT-PCR of *JAG1*, *DLL4* and *HEY1* after transduction of shRNA. (**B**) Representative phase contrast images (1 of 2 experiments) of the HUVEC network, after transduction of indicated shRNA and rescue by *EFNB2* analyzed by the Angiogenesis Analyzer from ImageJ. (**C**) Significantly changed measurements in the analyzed area: nodes surrounded by junctions (red dot surrounded by dark blue circle). Isolated elements (dark blue line). Segments (yellow).

*Figure 7 continued on next page*

*Figure 7 continued*

Number of branches (green) and meshes (light blue) were not significantly altered (not shown). Total segments length: sum of length of the segments. Number of pieces: sum of number of segments, isolated elements and branches detected. Total isolated branches length: sum of the length of the isolated elements. Red asterisks refer to comparisons between experimental and control situations. Black asterisks refer to comparisons between shRNA-mediated inhibition and LV-mediated rescue. Data are means ± s.d. ****$p<0.0001$; ***$p<0.001$; **$p<0.01$; *$p<0.05$, by ANOVA.

The online version of this article includes the following figure supplement(s) for figure 7:

**Figure supplement 1.** EPHRINB2 rescues defective capillary network formation after shRNA-mediated silencing of *EFNB2*.

mutants (*D'Amato et al., 2016*). In the later E14.5-E16.5 *Jag1^{flox}^;Pdgfb-iCre^{ERT2}^*, *Dll4^{flox}^;Pdgfb-iCre^{ERT2}^*, *Jag1^{flox}^;Cdh5-Cre^{ERT2}^* and *Dll4^{flox}^;Cdh5-Cre^{ERT2}^* mutant embryos, a thinner ventricular wall would be due to the lack of a well-formed coronary plexus (*Figure 8D*). At E12.5-E13-5, myocardial growth may also depend on 'angiocrine signals' from the un-perfused primitive coronary plexus (*Rafii et al., 2016*), as the diffusion limit of oxygen and nutrients from the endocardium is reached during the transition from endocardial to coronary myocardial perfusion. Therefore, ventricular compaction relies on two interconnected Notch-dependent processes: patterning and maturation of the chamber myocardium, and timely development of a functional coronary vessel network, as previously suggested (*D'Amato et al., 2016*). This may be clinically relevant to the study and treatment of cardiomyopathies.

# Materials and methods

## Key resources table

| Reagent type (species) or resource | Designation | Source or reference | Identifiers | Additional information |
|---|---|---|---|---|
| Genetic reagent | *Mus Musculus* (Mouse strain) | (*Nowotschin et al., 2013*) | CBF1:H2B-Venus | |
| Genetic reagent | *Mus Musculus* (Mouse strain) | (*Kisanuki et al., 2001*) | Tie2-Cre | |
| Genetic reagent | *Mus Musculus* (Mouse strain) | (*Wu et al., 2012*) | Nfatc1-Cre | |
| Genetic reagent | *Mus Musculus* (Mouse strain) | (*Wang et al., 2010*) | Pdgfb-iCre^{ERT2}^ | |
| Genetic reagent | *Mus Musculus* (Mouse strain) | (*Wang et al., 2010*) | Cdh5-Cre^{ERT2}^ | |
| Genetic reagent | *Mus Musculus* (Mouse strain) | (*Koch et al., 2008*) | Dll4^{flox}^ | |
| Genetic reagent | *Mus Musculus* (Mouse strain) | (*Mancini et al., 2005*) | Jag1^{flox}^ | |
| Genetic reagent | *Mus Musculus* (Mouse strain) | (*Grunwald et al., 2004*) | Efnb2^{flox}^ | |
| Genetic reagent | *Mus Musculus* (Mouse strain) | (*Radtke et al., 1999*) | Notch1^{flox}^ | |
| Genetic reagent | *Mus Musculus* (Mouse strain) | (*D'Amato et al., 2016*) | Mfng^{GOF}^ | |
| Sequenced-based reagent | FH1_DLL4 | 5'-GTTACACAGTGAAAAGCCAG-3' | KiCqStart_SIGMA qPCR primer | |
| Sequenced-based reagent | RH1_DLL4 | 5'-CTCTCCTCTGATATCAAACAC-3' | KiCqStart_SIGMA qPCR primer | |
| Sequenced-based reagent | FH1_JAG1 | 5'-ACTACTACTATGGCTTTGGC-3' | KiCqStart_SIGMA qPCR primer | |
| Sequenced-based reagent | RH1_JAG1 | 5'-ATAGCTCTGTTACATTCGGG-3' | KiCqStart_SIGMA qPCR primer | |
| Squenced-based reagent | FH1_HEY1 | 5'-CCGGATCAATAACAGTTTGTC -3' | KiCqStart_SIGMA qPCR primer | |

*Continued on next page*

*Continued*

| Reagent type (species) or resource | Designation | Source or reference | Identifiers | Additional information |
|---|---|---|---|---|
| Sequenced-based reagent | RH1_HEY1 | 5'-CTTTTTCTAGCTTAGCAGATCC-3' | KiCqStart_SIGMA qPCR primer | |
| Sequenced-based reagent | FH1_EFNB2 | 5'-AAAGTTGGACAAGATGCAAG-3' | KiCqStart_SIGMA qPCR primer | |
| Sequenced-based reagent | RH1_EFNB2 | 5'-TGTACCAGCTTCTAGTTCTG-3' | KiCqStart_SIGMA qPCR primer | |
| Transfected construct (human) | VSV-G | Viral Vectors Unit, CNIC, Spain | | Lentiviral construct to transfect shRNA |
| Cell line (include species here) | HUVEC | Lonza | | |
| Transfected construct (mouse) | pRLL-IRESeGFP | Addgene | | Lentiviral construct to transfect eGFP or Full-length murine EphrinB2 |
| Transfected construct (human) | shRNA to JAG1 | SIGMA | SHCLNG-NM_000214 Clone ID:NM_000214. 2-3357s21c1; Clone ID:NM_000214. 2-1686s21c1 | transfected construct (human) |
| Transfected construct (human) | shRNA to DLL4 | SIGMA | SHCLNV-NM_019074 Clone ID:NM_019074. 2-2149s21c1; Clone:NM_019074. 2-2276s21c1 | transfected construct (human) |
| Antibody | Anti-CD31/Pecam1 (Monoclonal Rat) | BD Biosciences Pharmingen | 550274 MEC13.3 | IF=1:100 |
| Antibody | Dll4 (Polyclonal Rabbit) | Santa Cruz Biotechnology | Sc-28915 | IF=1:100 |
| Antibody | Endomucin (Polyclonal Rat) | Santa Cruz Biotechnology | sc-65495 V.7C7 | IF=1:200 |
| Antibody | Jag1 (Monoclonal Rabbit) | Cell Signaling Technology | 2620 28H8 | IF=1:100 |
| Antibody | NFATc1 (Monoclonal Mouse) | Enzo Life Sciences | ALX-804-022-R100 7A6 | IF=1:100 |
| Antibody | Cleaved Notch1 (Val1744) (Monoclonal Rabbit) | Cell Signaling Technology | 4147 D3B8 | IF=1:100 |
| Antibody | p27 (Polyclonal Mouse) | Medical and Biological Laboratories | K0082-3 p27 Kip1 | IF=1:100 |
| Antibody | Notch 3 (Polyclonal Rabbit) | Abcam | ab23426 G00041 | IF=1:100 |
| Antibody | ERG (Monoclonal Rabbit) | Abcam | ab110639 EPR3863 | IF=1:100 |
| Antibody | Glut1 (Polyclonal Rabbit) | MERCK Millipore | 07-1401 C00222 | IF=1:100 |
| Antibody | BrdU (Monoclonal Mouse) | BD Biosciences | 347580 B-44 | IF=1:50 |
| Antibody | Isolectin IB4-Alexa Fluor 647 | Molecular Probes | I32450 | IF=1:300 |

*Continued on next page*

*Continued*

| Reagent type (species) or resource | Designation | Source or reference | Identifiers | Additional information |
|---|---|---|---|---|
| Antibody | Biotin Anti-rat | Vector laboratories | BA4001 | IF=1:200 |
| Antibody | Biotin Anti-rabbit | Jackson | 111-066-003 | IF=1:200 |
| Antibody | Anti-mouse Alexa Fluor 647 | Jackson | 115-606-003 | IF=1:200 |
| Antibody | Anti-mouse Alexa Fluor 488 | Invitrogen | A-11029 | IF=1:200 |
| Antibody | Anti-rat | Invitrogen | A-11006 | IF=1:200 |
| Peptide, recombinant protein | Human-Vegf165 | Human Peprotech | 100-20 | 10ng/ml |
| Commercial assay or kit | Click-iT EdU Imaging Kit | Thermo Fisher Scientific | C10340 | |
| Commercial assay or kit | HydroxyprobeTM-1 Plus Kit | Hydroxyprobe, Inc (HPI). | (Pimonidazole Hydrochloride CAS#70132-50-3) | |
| Other | Matrigel | Corning | 354234 | |
| Software | Cutadapt v1.6 | (*Martin, 2011*) | | |
| Software | RSEM v1.2.20 | (*Li and Dewey, 2011*) | | |
| Software | Limma | (*Ritchie et al., 2015*) | Bioconductor package | |
| Software | GOplot | (*Walter et al., 2015*) | | |
| Software | IPA | http://www.ingenuity.com | | |

## Mouse strains and genotyping

Animal studies were approved by the CNIC Animal Experimentation Ethics Committee and by the Madrid regional government (Ref. PROEX 118/15). All animal procedures conformed to EU Directive 2010/63EU and Recommendation 2007/526/EC regarding the protection of animals used for experimental and other scientific purposes, enforced in Spanish law under Real Decreto 1201/2005. Mouse strains were *CBF1:H2B*-Venus (**Nowotschin et al., 2013**), *Tie2-Cre* (**Kisanuki et al., 2001**), N*fatc1-Cre* (**Wu et al., 2012**), *Pdgfb-iCre^{ERT2}* (**Wang et al., 2010**), *Cdh5-Cre^{ERT2}* (**Wang et al., 2010**), *Dll4^{flox}* (**Koch et al., 2008**), *Jag1^{flox}* (**Mancini et al., 2005**), *Efnb2^{flox}* (**Grunwald et al., 2004**), *Notch1^{flox}* (**Radtke et al., 1999**), *Mfng^{GOF}* (**D'Amato et al., 2016**). To generate the *R26CAGDll4^{GOF}* transgenic line, a full-length mouse *Dll4* cDNA was obtained from clone IMAGE 6825525. The sequence was PCR amplified with Phusion High-Fidelity DNA Polymerase (NEB) and primers containing *BamHI* and *ClaI* sites and was cloned in-frame with 6 Myc Tag epitopes into the *BamHI* and *ClaI* sites of a *pCS2-MT* plamid. The *Dll4-MT* fragment was subcloned into pCDNA3.1 with *BamHI* and *EcoRI* and excised with *BamHI* and *NotI*. We then modified *pCCALL2* (**Lobe et al., 1999**) by cloning new *XbaI* sites before and after a CAG cassette, which includes a CMV enhancer/β-actin promoter and a rabbit β-globin polyA signal. The *KpnI* and *NotI* PCR *Dll4-MT* was cloned into the *BglII-NotI* sites of the modified *pCCALL2*. The *XbaI*-cassette containing *CAG-loxP-β-Geo-loxP-Dll4-IRESeGFP* was obtained by digestion and cloned into the *XbaI* site of the *pROSA26-1* plasmid (**Soriano, 1999**). The final construct was linearized with *XhoI* and electroporated into R1 mESCs derived from a cross of 129/Sv x 129/Sv-CP mice (**Nagy et al., 1993**). After G418 (200 μg/ml) selection for 7 days, 231 clones were picked. Homologous recombination was identified by Southern blot of *EcoRV*-digested DNA and hybridized with 5' and 3' probes. About 25% of the clones were positive, and we selected three clones to confirm karyotype. One positive clone was injected into C57/BL6C blastocysts to generate chimaeras that transmitted the transgene to their offspring. The resulting founders were genotyped by PCR of tail genomic DNA using primers targeting the *R26* locus before and after the cloning site and the transgene polyA signal.

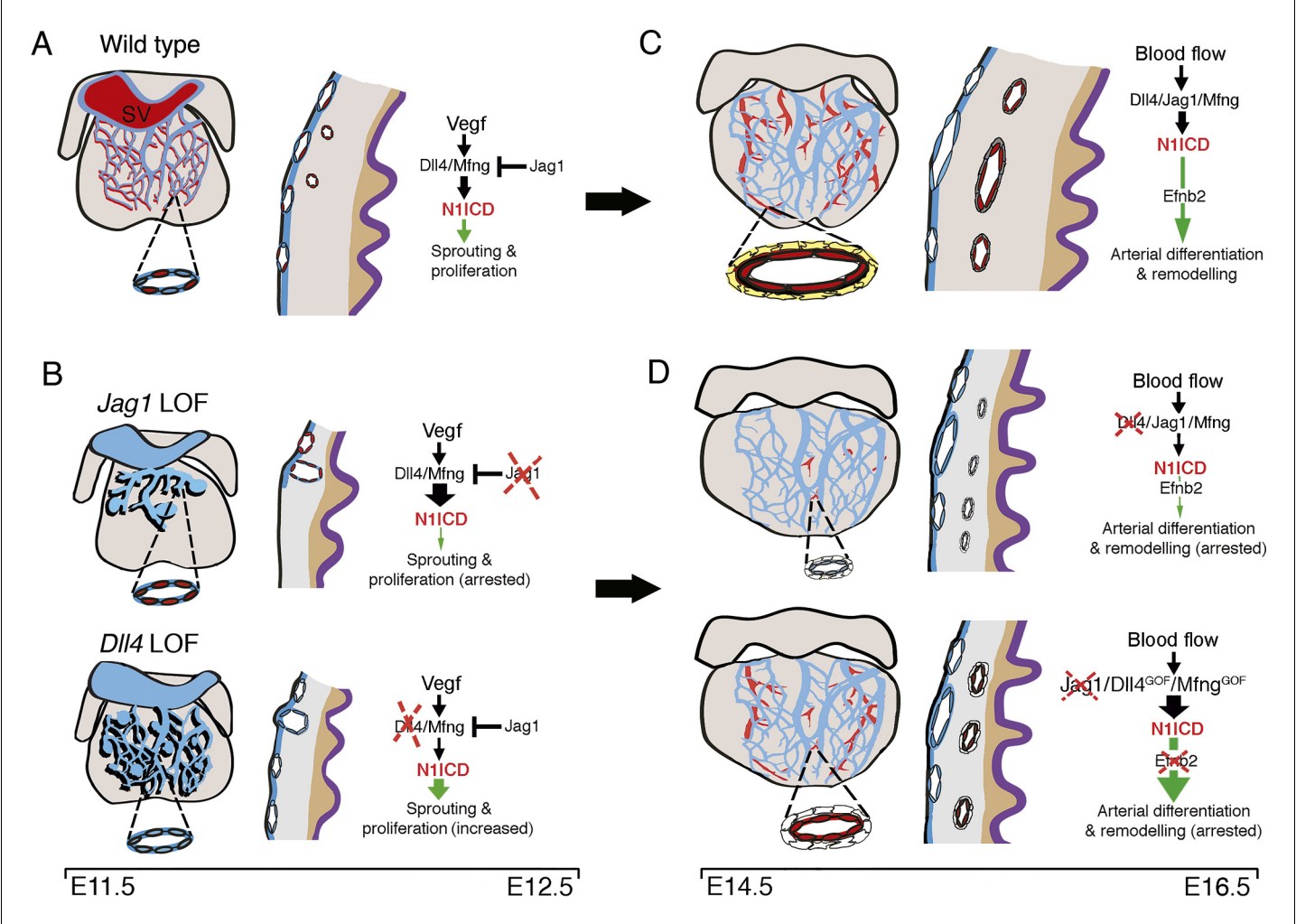

**Figure 8.** Coronary arterial development is regulated by a Dll4-Jag1-EphrinB2 signaling cascade. (**A,B**) Embryonic stages E11.5-E12.5. Development of the primitive coronary plexus. (**A**) Left: sinus venous (SV)-derived vessels covering the dorsal side of a wild-type heart. Zoomed view showing a subepicardial vessel with a subset of ECs expressing N1ICD (red). These N1ICD-expressing cells are equivalent to pre-artery cells defined by *Su et al. (2018)*. Center: cross-section through the ventricular wall. SV-derived ECs invade the sub-epicardial space over the myocardium to cover the heart dorsally. Some pre-artery ECs invade the myocardium and begin to differentiate into arteries. Right: during EC sprouting and proliferation, a regulatory balance between Dll4/Mfng and Jag1 modulates Notch signaling output downstream of Vegf. (**B**) Left: E12.5 *Jag1^flox^;Nfatc1-Cre* (Jag1LOF; top) exhibit arrested coronary angiogenesis, *Dll4^flox^;Nfatc1-Cre* mutants (Dll4LOF; bottom) display increased angiogenesis, and both mutants display vascular malformations. The zoomed views show details of a sub-epicardial vessels with increased N1ICD expression in the *Jag1*LOF mutants (red), and decreased expression in the *Dll4*LOF mutants (blue). Center: the sub-epicardial capillary plexus is either poorly developed in *Jag1*LOF mutants or over-developed in *Dll4*LOF mutants, and with vascular malformations. Mutant hearts have a thin myocardial wall, but maintain myocardial patterning (grey-brown boundary). Right: Endocardial *Jag1* deletion disrupts the regulatory balance with Dll4/Mfng, leading to increased N1ICD and arrested migration and proliferation. In contrast, endocardial *Dll4* deletion results in decreased N1ICD signaling, and increased network complexity and proliferation. (**C,D**) Embryonic stages E14.5-E16.5. Arterial differentiation and plexus remodeling. (**C**) Left: coronary vasculature at the dorsal aspect of the wild-type heart. Zoomed view showing a coronary artery with an inner layer of ECs expressing N1ICD (red) and an outer layer of smooth muscle cells (yellow). Center: capillary differentiation and patterning give rise to large veins sub-epicardially and large arteries intra-myocardially. Right: systemic blood flow activates Dll4/Jag1/Mfng/N1ICD/EphrinB2 signaling to drive terminal arterial differentiation and remodeling. (**D**) Left: in coronary endothelial-*Dll4^flox^;Pdgfb-iCre^ERT2^* or *-Jag1^flox^;Pdgfb-iCre^ERT2^* mutants, the coronary vasculature is mis-patterned. Zoomed view detailing near absence of arteries in *Dll4* mutants, and comparatively smaller caliber coronary artery in *Jag1* mutants. The arteries in both mutants have an inner layer of a 'leaky' endothelium and an outer layer of poorly differentiated perivascular cells (white). Center: *Dll4* mutants are characterized by the near-absence of coronary arteries while arteries in *Jag1* mutants are decreased. In either case, veins are unaffected. Mutant hearts have a thin myocardial wall but maintain myocardial patterning (grey-brown boundary). Right: Coronary endothelial-*Jag1* or *Dll4*LOF leads to increased or decreased N1ICD respectively, causing arrested arterial differentiation and remodeling. Inactivation of *Efnb2* leads to a similar phenotype. Endothelial Notch gain-of-function (GOF), resulting from *Dll4* or *Mfng* overexpression, also leads to increased N1ICD and disrupted arterial differentiation and remodeling.

Throughout the MS, *Jag1*<sup>flox/flox</sup>;*Nfatc1-Cre* mice are called *Jag1*<sup>flox</sup>;*Nfatc1-Cre* for simplicity; this also applies to the *Dll4*<sup>flox</sup>;*Nfatc1-Cre* mice, *Jag1*<sup>flox</sup>;*Pdgfb-iCre*<sup>ERT2</sup> mice, *Dll4*<sup>flox</sup>;*Pdgfb-iCre*<sup>ERT2</sup>, *Dll4*<sup>flox</sup>;*Cdh5-Cre*<sup>ERTT2</sup>, *Jag1*<sup>flox</sup>;*Cdh5-Cre*<sup>ERT2</sup> mice, *Dll4*<sup>flox</sup>;*Cdh5-Cre*<sup>ERT2</sup>, *Dll4*<sup>flox</sup>;*Cdh5-Cre*<sup>ERTT2</sup> *Mfng*<sup>GOF</sup>;*Tie2-Cre* mice (which carry two extra copies of *Mfng*), and *Efnb2*<sup>flox</sup>;*Nfatc1-Cre* mice, but not to *Dll4*<sup>GOF</sup>;*Tie2-Cre* mice, which carry only one extra copy of *Dll4*.

## Tamoxifen induction

Double heterozygous *Cdh5-Cre*<sup>ERTT2/+</sup>;*Jag1*<sup>flox/+</sup> females were crossed with homozygous *Jag1*<sup>flox/flox</sup> males, and pregnant females were administered by oral gavage 200 µl of tamoxifen solution (Sigma, T5648; 5 mg/ ml; generated by diluting 50 mg of tamoxifen in ml of 95% ethanol plus 9 ml of Corn Oil). To obtain *Jag1*<sup>flox/flox</sup>;*Cdh5-Cre*<sup>ERTT2</sup> embryos, pregnant females were induced at E9.5 and opened at E16.5. To obtain *Dll4*<sup>flox/flox</sup>;*Cdh5-Cre*<sup>ERTT2</sup> embryos, *Dll4*<sup>flox/flox</sup> males were crossed with double heterozygous *Cdh5-Cre*<sup>ERTT2/+</sup>;*Dll4*<sup>flox /+</sup> females, and pregnant females were induced at E12.5 and opened at E15.5. With *Pdgfb-iCre*<sup>ERT2</sup>, double heterozygous *Pdgfb-iCre*<sup>ERT2/+</sup>;*Jag1*<sup>flox/+</sup>(or *Dll4*<sup>flox/+</sup>) males were crossed with homozygous *Jag1*<sup>flox/flox</sup> or *Dll4*<sup>flox/flox</sup> females. Pregnant females were induced at E12.5 and dissected at E15.5 (for *Jag1*<sup>flox</sup>), or induced at E14.5 and dissected at E15.5 for *Dll4*<sup>flox</sup>.

## Histology and in situ hybridization

Hematoxylin and eosin (H & E) staining and in situ hybridization (ISH) on sections were performed as described (*Kanzler et al., 1998*). Details of probes will be provided on request.

## Whole-mount immunofluorescence

Embryos were dissected and fixed for 2 hr in 4% paraformaldehyde (PFA) in PBS at 4°C, then permeabilized for 1 hr with 0.5% Triton X-100/PBS and subsequently blocked for 1 hr in Histoblock solution (5% goat serum, 3% BSA, 0.3% Tween-20 in PBS). After several washes in PBS-T (PBS containing 0.1% Tween-20), embryos were mounted in 1% agar in a 60 mm petri dish. For whole-mount immunofluorescence on E11.5-E12.5 *CBF:H2B-Venus* embryos endogenous GFP, was imaged with a NIKON A1R confocal microscope. Z-stacks were captured every 5 µm.

For EdU immunofluorescence, pregnant females at E12.5 were injected intraperitoneally with 100 µl EdU nucleotides (2 mg/ml in PBS). The mice were euthanized 1 hr later and fixed in 4% PFA. EdU incorporation was detected with the Click-iT EdU Imaging Kit (Thermo Fisher Scientific, C10340) according to manufacturer's instructions. For IsoB4 and Endomucin immunostaining on E11-4-E16.5 embryos, imaging by confocal z-projection of the deeper section of the myocardium captures the arteries (IsoB4-stained) with the outer section corresponding to the veins (IsoB4+Endomucin-stained). N1ICD embryos were fixed for 5 hr at 4°C in frozen Methanol, washed and incubated 20 min at 98°C in DAKO target retrieval solution pH6 (Agilent). After washing in H$_2$O, hearts were fixed again in acetone at −20°C for 10 min. After washing in PBT, the hearts were blocked in 10% FBS, 5% BSA, 0.4% TritonX100 for 3 hr at RT under gentle rocking. Hearts were incubated with primary N1ICD-antibody at 1:500 for 2 days at 4°C and allowed to settle at RT for 1 hr. This was followed by washing 5 hr in PBS/0.4% TritonX100 at RT. Hearts were then incubated in presence of anti-rabbit HRP (1:500), DAPI (1:1500) and anti-VE cadherin (1:500) overnight at 4°C. After resting for 1 hr at RT, hearts were washed in PBT, and 30 min in presence TSA (tyramides) diluted 1:200 in PBS. Quantifications were carried out using ImageJ software. The main coronary trees were selected with the tool 'Freehand selection' manually and measured directly from the autoscaled images obtained by Z-projection. The selected area was quantified in µm$^2$. Confocal microscopy analysis was carried out on a Nikon A1R or Leica LAS-AF 2.7.3.

## Immunofluorescence on sections

Paraffin sections (7 µm) were incubated overnight with primary antibodies, followed by 1 hr incubation with a fluorescent-dye–conjugated secondary antibody. N1ICD, Dll4, Jag1, and p27 were immunostained using tyramide signal amplification (TSA) (*Del Monte et al., 2007*); see *Supplementary file 4* for antibodies. All immunostainings were performed in the same way except for co-immunostaining of SM22a (+ secondary chicken anti-goat-594) and Notch3 (+secondary donakey anti-rabbit-488), which required continued permeabilization in PBS 1X + 0.3% TritonX100.

Images were processed using ImageJ software. For p21 and N1ICD quantification, the number of positive nuclei was divided by the total number of nuclei counted on sections ($\geq$3). For αSMA and Notch3 quantification, the number of mural cells surrounding intramyocardial vessels expressing both SMA+ and Notch3+ (double positives) was counted and divided by the total number of intra-myocardial vessels defined by Iso B4 staining in the same section. For SM22a and Notch3 quantification, the number of double- SM22 and Notch3 positive cells was divided by the total number Notch3-positive cells counted on sections ($\geq$3).

## Hypoxia analysis

Pimonidazole hydrochloride (Hypoxyprobe) was injected intraperitoneally at 60 mg/kg to E12.5 pregnant females. After 2.5 hr, embryos were dissected in cold PBS and fixed O/N in 4% PFA. Embryos were paraffin embedded. Sections at 7 μm were deparaffinized and boiled in 10 mM, pH = 6 Na Citrate solution. For immunofluoresence, sections were permeabilized with 0.4% Triton in PBS, blocked with 5% FBS and incubated anti-Glut1 (1:100) and anti-Pymonidazole (1:200) antibodies O/N at 4°C. Secondary antibodies (anti-Rabbit Alexa568 for Glut1 and anti-mouse biotinylated for Pymonidazole 1:400, respectively) were incubated for 1 hr at RT. Pymonidazole signal was amplified using Fluorescein Tyramides (1:100). Counterstaining: IB4 (endocardium/endothelium) 1:200; DAPI (nuclei) 1:3000.

## Mouse ventricular explant culture and immunofluorescence

Heart ventricular explants were performed as previously described with minor modifications (*Wu et al., 2012*). Ventricles were dissected from E10.5 or E11.5 embryos (with removal of the out-flow tract and atria), rinsed with PBS to remove circulating cells, and placed in Nunc four-well plates. Matrigel (Corning Matrigel Basement Membrane Matrix, *LDEV-Free, 10 mL growth factor reduced, BD Biosciences 354234) was diluted 1:1 with DMEM plus 10% FBS and 10 ng/ml Vegf (Human-Vegf 165 Peprotech). Each well had a total volume of 400 μl, and 3–4 hearts were cultured for 6 days and then fixed in 4% PFA4% 10 min and washed twice with PBS1X for 15 min. Explants were permeabilized with 0.5% Triton X-100 for 1 hr, blocked with Histoblock solution (FBS Histoblock) for 2–3 hr at RT, and incubated with anti-CD31 1:100 (Purified Rat Anti-Mouse CD31 BD Biosciences Pharmingen) overnight at 4°C. Anti-rat biotinylated was used as secondary antibody and diluted in BSA 1:150. Staining was amplified with the ABC kit and 3 min of TSA (tyramides) diluted 1:100 in PBS.

## Quantification of compact myocardium thickness

The method used was a modification of that described by *Chen et al. (2009)*; *Yang et al. (2012)*. Briefly, 7 μm paraffin sections from E12.5, E15.5 and E16.5 wild type (WT) and mutant hearts were stained with anti-CD31 or Endomucin (Emcn) and anti-MF20 antibodies to visualize ventricular structures. Confocal images were obtained with a NIKON A1R confocal microscope. Measurements were made using ImageJ software. In E15.5 and E16.5 heart sections, endocardial cells were stained with anti-Emcn and myocardium with anti-cTnT. Left and right ventricles were analyzed separately. For each measurement, settings were kept constant for all images, using the scale bar recorded in each image as the reference distance. The thickness of the compact myocardium was measured by dividing the ventricle into left and right parts. Several measurements were taken in each region and the mean was expressed in μm.

## Quantification of *Jag1* and *Dll4* expression in sections

Quantification of ISH signal were carried out using ImageJ software. Compact myocardium area was selected manually with the tool 'Polygon selection'. Coronary area was identified in this selection by color intensity using 'Color Threshold plugin' with the settings: Hue: Min = 145 Max = 199, Saturation: Min = 60 Max = 255, Brightness: Min = 0 Max = 255. Positive coronary area for *Dll4* or *Jag1* expression was calculated dividing the coronary area positive for each gene by the total compact myocardium area. Statistical analysis was assessed by Student's t-test.

## India ink perfusion

Embryos were collected from E15.5 to E18.5 in PBS containing 2 μg/ml heparin. The thoracic cavities were immediately cut and transferred to DMEM containing 10%FBS and heparin on ice. Hearts were

carefully dissected from the surrounding tissue and then placed on an inverted Petri dish and gently dried to avoid movement during perfusion. India ink or red tempera was diluted in PSB/heparin and injected from the ascending aorta with a borosilicate glass tube thinned to the appropriate diameter and attached to a mouth pipette. India ink was slowly injected by minute puffs of breath during diastolic intervals. Hearts were fixed in 4% PFA, dehydrated, cleared in BABB (benzyl alcohol: benzyl benzoate, 1:1) and imaged with a stereomicroscope.

## Lentiviral production

Bacterial glycerol stocks for JAG1 and DLL4 MISSION shRNA were purchased from SIGMA. MISSION shRNAs: shRNA JAGGED1_2: clone ID: NM_000214.2–3357 s21c1; shRNA JAGGED1_3: clone ID: NM_000214.2–1686 s21c1; shRNA DLL4_1: clone ID_ NM_019074.2–2149 s21c1; shRNA DLL4_1: clone ID_ NM_019074.2–821 s21c1. Concentrated lentiviral particles were produced by triple transfection in HEK293T cells. Briefly, HEK293T cells were cultured in 150 mm plates for transfection of the lentiviral vectors. The psPAX2 packing plasmid, the pMD.G envelope plasmid coding for the VSV-G glycoprotein (Viral Vectors Unit, CNIC, Spain) and the LTR-bearing shuttle lentiviral plasmids were cotransfected using the calcium phosphate method. The transfection medium was replaced with fresh medium 16 hr post-transfection. Viral supernatants were harvested at 72 hr post-transfection, filtered through a 0.45 µm filter (Steriflip-HV, Millipore, MA) and concentrated by ultracentrifugation. Precipitated viruses were resuspended in pre-chilled 1X PBS, aliquoted and stored at −80°C for further use. Viral titer was measured on viral genomes by qPCR using the standard curve method with primers against the LTRs. Forward primer: 5'-AGCTTGCCTTGAGTGCTTCAA-3'; Reverse primer: 5'-AGGGTCTGAGGGATCTCTAGTTA-3'. Full-length murine *EphrinB2 (Efnb2)*, kindly provided by Ralf Adams (MPI for Molecular Biomedicine, University of Münster, Germany), was subcloned into the lentiviral vector pRLL-IRES-eGFP (Addgene). Concentrated lentiviruses expressing pRLL-IRES-eGFP or pRLL-Mfng-IRES-eGFP were obtained as described (*Esteban et al., 2011*). Viruses were titrated in Jurkat cells, and infection efficiency (GFP-expressing cells) and cell death (propidium iodide staining) were monitored by flow cytometry.

## Culture and infection on HUVEC

Human umbilical vein endothelial cells (HUVECs) were maintained in supplemented EGM2 medium (EGM2 Bulletkit, K3CC-3162 Lonza). Cells were transduced on suspension at a multiplicity of infection (m.o.i.) of 80 with combinations of 2 shRNAs against JAG1 or DLL4 together with GFP or EFNB2 overexpressing lentivirus and seeded onto 24-well plates or 12-well plates at a density of $3 \times 10^4$ or $6 \times 10^4$ cells/well respectively. Infection efficiency was near 100%. HUVECs were harvested after 48 hr. Gene expression analysis was performed using KiCqStart SYBR Green predesigned primers (Sigma-Aldrich).

## Angiogenesis assays on matrigel

Angiogenesis in vitro was investigated using endothelial cell tube formation assay on Matrigel (#354234 Corning) as previously described (*Grant et al., 1991*). Briefly 50 µl of ice cold Matrigel was coated on a 96 micro-well plate as a base for tube formation. After allowing the gel to settle in the incubator for 30 min at 37°C, 5% $CO_2$, HUVECs in basic EGM medium (EBM2 H3CC-3156, Lonza) were seeded in triplicate at a density of $12.5 \times 10^4$ on Matrigel and incubated at 37°C, 5% $CO_2$. After 5 hr, tube formation was monitored by phase-contrast microscope. The tube networks were quantified using NIH Image J with Angiogenesis plugin software (http://image.bio.methods.free.fr/ImageJ/?Angiogenesis-Analyzer-for-ImageJ). Excel data were represented on Graphpad Prism seven as mean with SD. For statistical analysis a one-way ANOVA and Tukey post-hoc tests were performed.

## Statistical analysis

Statistical analysis was carried out using Prism 7 (GraphPad). All statistical tests were performed using two-sided, unpaired Student's t-tests, except for *Figures 1*, *2* and *7* and figures Supplements 4, 9 and 12 where we performed one-way ANOVA to assess statistical significance with a 95% confidence interval, where numerical data are presented as mean ± SD; results are marked with one asterisk (*) if $p < 0.05$, two (**) if $p < 0.01$, and three (***) if $p < 0.001$. Sample size was chosen empirically

according to previous experience in the calculation of experimental variability. No statistical method was used to predetermine sample size. All experiments were carried out with at least three biological replicates. The numbers of animals used are described in the corresponding figure legends. Animals were genotyped before the experiment and were caged together and treated in the same way. Variance was comparable between groups throughout the manuscript. We chose the appropriate tests according to the data distributions. The experiments were not randomized. The investigators were not blinded to allocation during experiments and outcome assessment.

### RNA-sequencing

RNA was isolated at E12.5, from whole hearts of *Jag1*$^{flox}$;*Nfatc1-Cre* and *Dll4*$^{flox}$;*Nfatc1-Cre* embryos, as well as their WT counterparts (four replicate samples for each condition, pooling four embryos per sample, in all cases). At E15.5, RNA was isolated from ventricles of *Dll4*$^{flox}$;*Cdh5-Cre*$^{ERT2}$ (three replicate samples, pooling three embryos per sample), as well as their WT counterparts (four replicate samples, pooling three embryos per sample). RNA-Seq data for *Jag1*$^{flox}$;*Nfatc1-Cre*, *Dll4*$^{flox}$;*Nfatc1-Cre*, and *Dll4*$^{flox}$;*Cdh5-Cre*$^{ERT2}$ experiments was generated by CNIC's Genomics Unit. RNA-Seq sequencing reads were pre-processed by means of a pipeline that used FastQC (http://www.bioinformatics.babraham.ac.uk/projects/fastqc/) to assess read quality and Cutadapt v1.6 (*Martin, 2011*) to trim sequencing reads, thus eliminating Illumina adaptor remains, and to discard reads shorter than 30 bp. Resulting reads were mapped against the mouse transcriptome (GRCm38 assembly, Ensembl release 76) and quantified using RSEM v1.2.20 (*Li and Dewey, 2011*). Around 80–90% of the reads participated in at least one reported alignment. Expected expression counts calculated with RSEM were then processed with an analysis pipeline that used Bioconductor package Limma (*Ritchie et al., 2015*) for normalization and differential expression testing in six pairwise contrasts involving mutant versus WT comparisons. We discarded two of the *Dll4*$^{flox}$;*Nfatc1-Cre* samples based on preliminary analyses of *Dll4* expression levels and clustering patterns in diagnostic principal component analysis plots. Changes in gene expression were considered significant if associated with a Benjamini-Hochberg (BH) adjusted p-value<0.05. The number of differentially expressed genes detected in comparisons between *Jag1*$^{flox}$;*Nfatc1-Cre*, *Dll4*$^{flox}$;*Nfatc1-Cre* and *Dll4*$^{flox}$;*Cdh5-Cre*$^{ERT2}$ and their control counterparts was 205, 257 and 156, respectively. Enrichment analyzes were performed with IPA (Ingenuity Pathway Analysis, Qiagen, http://www.ingenuity.com). IPA was used to identify collections of genes associated with canonical pathways, common upstream regulators, or functional terms significantly overrepresented in the sets of differentially expressed genes; statistical significance was defined by Benjamini-Hochberg adjusted p-value<0.05. Circular plots summarizing the association between genes and enriched pathways, upstream regulators, and functional terms were generated with GOplot (*Walter et al., 2015*).

## Acknowledgements

We thank the CNIC Genomics Unit for RNA-seq experiments and S Bartlett (CNIC) for English editing. This study was supported by grants SAF2016-78370-R, CB16/11/00399 (CIBER CV), and RD16/0011/0021 (TERCEL) from the Spanish Ministry of Science, Innovation and Universities (MCIU) and grants from the Fundación BBVA (Ref.: BIO14_298) and Fundación La Marató (Ref.: 20153431) to JLDLP. JG-B is funded by Atracción de Talento Program from the Comunidad de Madrid. The cost of this publication was supported in part with funds from the ERDF. The CNIC is supported by the MCIU and the Pro-CNIC Foundation and is a Severo Ochoa Center of Excellence (SEV-2015–0505).

## Additional information

### Funding

| Funder | Grant reference number | Author |
| --- | --- | --- |
| Spanish Ministry of Science, Innovation and Universities | SAF2016-78370-R | José Luis de la Pompa |
| Spanish Ministry of Science, Innovation and Universities | CB16/11/00399 | José Luis de la Pompa |

| Spanish Ministry of Science, Innovation and Universities | RD16/0011/0021 | José Luis de la Pompa |
| Fundación BBVA | BIO14_298 | José Luis de la Pompa |
| Fundació la Marató de TV3 | 20153431 | José Luis de la Pompa |
| Comunidad de Madrid | Atracción de Talento Program | Joaquim Grego-Bessa |

The funders had no role in study design, data collection and interpretation, or the decision to submit the work for publication.

### Author contributions

Stanislao Igor Travisano, Investigation, Visualization; Vera Lucia Oliveira, Investigation, Visualization, Methodology; Belén Prados, Validation, Investigation; Joaquim Grego-Bessa, Validation, Investigation, Visualization; Rebeca Piñeiro-Sabarís, Investigation, Methodology; Vanesa Bou, Investigation; Manuel J Gómez, Software, Validation, Investigation; Fátima Sánchez-Cabo, Software, Supervision, Validation, Investigation; Donal MacGrogan, Data curation, Formal analysis, Supervision, Investigation; José Luis de la Pompa, Resources, Supervision, Funding acquisition, Visualization, Project administration

### Author ORCIDs

Joaquim Grego-Bessa (ID) https://orcid.org/0000-0002-0938-2346
Donal MacGrogan (ID) http://orcid.org/0000-0003-2808-8422
José Luis de la Pompa (ID) https://orcid.org/0000-0001-6761-7265

### Ethics

Animal experimentation: Animal studies were approved by the CNIC Animal Experimentation Ethics Committee and by the Madrid regional government (Ref. PROEX 118/15). All animal procedures conformed to EU Directive 2010/63EU and Recommendation 2007/526/EC regarding the protection of animals used for experimental and other scientific purposes, enforced in Spanish law under Real Decreto 1201/2005.

### Decision letter and Author response

Decision letter https://doi.org/10.7554/eLife.49977.sa1
Author response https://doi.org/10.7554/eLife.49977.sa2

# Additional files

### Supplementary files

• Source data 1. Numerical data that are represented as a graph in a Figure or figure supplement, distributed in sheets corresponding to a given Figure or figure supplement by their order of appearance in the text, where they are also referred to.

• Supplementary file 1. Lethality phases of $Jag1^{flox}$;Nfatc1-Cre, $Dll4^{flox}$;Nfatc1-Cre, $Jag1^{flox}$;Pdgfb-iCre$^{ERT2}$, $Dll4^{flox}$;Cdh5-iCre$^{ERT2}$ $Dll4^{flox}$;Cdh5-Cre$^{ERT2}$, $Mfng^{Gof}$;Tie2-Cre, $Efnb2^{flox}$;Nfatc1-Cre embryos.

• Supplementary file 2. Lists of differentially expressed genes identified after analysis of RNA-seq data, for contrasts between E12.5 $Jag1^{flox}$;Nfatc1-Cre (Jag1Nfatc1 211 Deg); E12.5 $Dll4^{flox}$;Nfatc1-Cre (Dll4Nfatc1 274 Deg); E16.5 $Dll4^{flox}$;Cdh5-Cre$^{ERT2}$ (Dll4Cdh5 163 Deg) and their control counterparts. Upregulated genes appear in red background and downregulated genes in blue.

• Supplementary file 3. Results obtained with IPA for the sets of genes detected as differentially expressed (adjusted $P$ value < 0.05) in contrasts between $Jag1^{flox}$;Nfatc1-Cre (JAG1NF), $Dll4^{flox}$;Nfatc1-Cre (Dll4NF), $Dll4^{flox}$;Cdh5-Cre$^{ERT2}$ (Dll4Cdh5) and their control counterparts. Upstream Regulator analyses results, in sheets labelled as 'UpRegTransc', are restricted to molecules of type 'transcription regulator'. For each predicted upstream regulator, tables describe expression log ratio (when available), predicted activation state, activation z-score, enrichment p_value and the collection

of differentially expressed genes that are targets of the regulator. Downstream Effect analyses results, in sheets labelled as 'CardSystDevFunc', are restricted to functions of category 'Cardiovascular System Development and Function'. For each functional term, tables describe enrichment p value, predicted activation state, activation z-score and the collection of differentially expressed genes associated to that particular function. Positive and negative z-score values suggest activation or inhibition of the corresponding upstream regulator or function in the mutant or control condition, respectively; abs(z-score)>2 and p value < 0.05 are considered significant.

• Supplementary file 4. List of primary and secondary antibodies used in this study to immunodetect proteins in whole-mount or paraffin sections.

• Transparent reporting form

## Data availability

Sequencing data have been deposited in GEO under accession code GSE110614.

The following dataset was generated:

| Author(s) | Year | Dataset title | Dataset URL | Database and Identifier |
|---|---|---|---|---|
| Travisano SI, Mac-Grogan D, de la Pompa JL | 2018 | Coronary arterial development is regulated by a Dll4-Jag1-EphrinB2 signaling cascade | https://www.ncbi.nlm.nih.gov/geo/query/acc.cgi?acc=GSE110614 | NCBI Gene Expression Omnibus, GSE110614 |

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
