## [Decision Letter]

Thank you for submitting your article "Coronary arterial development is regulated by a Dll4-Jag1-EphrinB2 signaling cascade" for consideration by *eLife*. Your article has been reviewed by three peer reviewers, including Bin Zhou as the Reviewing Editor and Reviewer #1, and the evaluation has been overseen by Didier Stainier as the Senior Editor. The following individual involved in review of your submission has agreed to reveal their identity: Nicola Smart (Reviewer #3).

The reviewers have discussed the reviews with one another and the Reviewing Editor has drafted this decision to help you prepare a revised submission.

Summary:

Unraveling the molecular mechanism regulating coronary vessel formation provides important information to neovascularization in coronary artery diseases. In this study, Travasiano et al. used mouse genetic Cre-loxP system as well as in vitro culture experiments to determine whether Notch ligands *Dll4* and *Jag1* promote or attenuate coronary angiogenesis. Their data showed that sinus venosus endocardial knockout of *Jag1* inhibits coronary sprouting from sinus venosus, while *Dll4* removal promotes excessive coronary sprouting, indicating the ligands antagonism in regulating coronary angiogenesis. Gain of function by *Dll4* or glycosyltransferase MFng over expression inhibit coronary artery formation. Authors then linked Notch signaling to downstream target EphrinB2, which mediates arterial differentiation. While Notch signaling in angiogenesis regulation and artery formation has been reported in other vascular beds (e.g. retina vascular plexus), the role in coronary angiogenesis and artery maturation has not been reported and remains incompletely understood currently. This study, therefore, offers a sufficient advance to the field of coronary vascular development and may provide new insight to coronary neovascularization in diseases such as myocardial infarction. Overall, this finding is interesting and important, and the major conclusion is supported by the data presented.

Essential revisions:

1) Deletion efficiency of genes by in situ or immunostaining detection. The authors claimed that Late endothelial *Jag1* or *Dll4* inactivation disrupts coronary plexus remodeling. The authors should show the relative target gene *Dll4, Jag1* deletion efficiency in inducible mice, by immunostaining or in-situ?

2) A schematic model need to be provided. Since *Jag1* and *Dll4* have opposing effects on Notch signaling, based on the current data, how do the authors propose these ligands and Notch signaling shape the pre-artery fate and coronary vessel development temporally and spatially? Since phenotypes were analyzed at different stages during coronary plexus formation, arterial differentiation and myocardial growth, the authors could consider to provide a schematic model to help with the readability.

3) More detailed in situ hypoxia measurement need to be included. It is difficult to understand why hypoxia should manifest as early as E12.5 in the *Dll4/ Jag1* mutants, if, as inferred, this is due to malformed coronary plexus and perfusion defects. At this stage, SV and endocardial sprouting is not very advanced (in control hearts) and the mutant hearts are less compacted, thus less reliant on vessels for perfusion. Perhaps this could be reconciled if the authors assessed hypoxia in situ e.g. with hypoxyprobe (or at the very least, Glut1 across the whole heart) to correlate regional hypoxia with coronary vessel sprouting. If, instead, this is a stress response to the loss of angiocrine stimuli, as alluded, the authors should validate some putative factors from the RNA-Seq data to support this conclusion.

4) Other more specific markers for smooth muscle should be used. αSMA is used to label smooth muscle cells, however, it is expressed more broadly in mesenchymal cells and cardiomyocytes throughout the embryonic heart prior to ~E16.5 and this is reflected in the images. The authors should attempt to stain for SM22a which, although similarly non-specific, shows better enrichment in VSMCs, relative to the other cell types. Improved visualization may provide further insight into the nature of the defect e.g. are epicardium-derived precursors trapped or poorly differentiated?

Reviewer #1:

Understanding how coronary arteries are formed is important for developmental biology and also postnatal coronary neovascularization after cardiac injuries. In this study, the authors reported that coronary arterial precursors are specified in the SV prior to the formation of primary coronary plexus, while Dll4-Jag1-EphrinB2 signaling mediated subsequent arterial differentiation. Inactivation of *Dll4* increase capillary growth while *Jag1* removal blocks sprouting. Deletion of arterial differentiation target Efnb2 causes coronary arterial defects. This finding is interesting and important to the field. Overall, the conclusion is supported by the data presented. I have a few points for author to consider and clarify.

The ligand antagonism mechanism proposed by author for coronary angiogenesis regulation is interesting. Is this *Jag1/Dll4* antagonism restricted to coronary development or has already been known in other organotypic vessel development?

Deletion of *Dll4* causes excessive expansion of the primitive coronary vessels and at the same time has thin (half) myocardium in the mutant. This is an interesting mutant phenotype as it separates myocardial growth and regulation of coronary angiogenesis. Could author elaborate more on this and also study how the angiogenesis factors such as VEGF in thin myocardium regulates coronary vessels?

The authors claimed that Late endothelial *Jag1* or *Dll4* inactivation disrupts coronary plexus remodeling. The authors should show the relative target gene *Dll4, Jag1* deletion efficiency in inducible mice, by immunostaining or in-situ?

Does *Pdgfb-iCre^ERT2^* also target endocardial cells in the heart in addition to coronary vascular endothelial cells?

There is reduced artery coverage in Efnb2 knockout hearts. How about the pre-artery cells (Su et al., 2018) in the mutant heart? How about pericyte/smooth muscle cells recruitment in the mutant?

The authors described in the manuscript that p27 nuclear staining was increased 2-fold which indicate decreased cell proliferation, so in which cell p27 increased expression level, cardiomyocytes or endothelial?

The authors should pay more attention to the details in the manuscript and figures. Such as how the EDU inject in the experiments? please describe it. More detailed information for experiments should be provided in the Materials and methods section.

Reviewer #2:

Travisano and colleagues performed a comprehensive analysis of the roles of Notch ligands in coronary vessel development by manipulating Jag-1 and *Dll4* using combinations of various Cre-floxed lines. Using both gain and loss of function analysis of Notch signaling pathway components and both in vivo and in vivo (including a novel ventricular explant) assays, the authors showed that Notch signaling is essential for sinus venosus sprouting angiogenesis for coronary plexus formation, coronary vessel maturation and artery formation. The authors also linked Notch functions to a known downstream target, EphrinB2, which mediates angiogenesis and arterial differentiation. Overall, the authors have done extensive phenotypic analyses and provided data of excellent quality. The manuscript is also very well written. The manuscript can be improved by providing more specific mechanistic insights of certain phenotypes. However, considering that the work covered in this manuscript has filled in a missing gap in our knowledge in Notch signaling in coronary vessel development and provided entry points for more future investigations, it warrants publication in *eLife*.

1) The overall finding here is consistent with the known roles of Notch signaling. Since *Jag1* and *Dll4* have opposing effects on Notch signaling, based on the current data, how do the authors propose these ligands and Notch signaling shape the pre-artery fate and coronary vessel development temporally and spatially? Since phenotypes were analyzed at different stages during coronary plexus formation, arterial differentiation and myocardial growth, the authors could consider to provide a schematic model to help with the readability.

2) Figure 6 legend, the authors should clarify the labels for the vessel caliber, branch point distance and the endothelial number (i.e. what are labeled by arrowheads, asterisks and arrows?). Since these features seem to vary in different areas of the explants, the authors should label these in all magnified panels.

Reviewer #3:

The study by Travasiano et al. examined the requirement of the Notch pathway in development of the coronary vasculature, specifically to determine whether the ligands *Jag1* and *Dll4* promote and attenuate angiogenesis, respectively, from the main embryonic sources (sinus venosus and endocardium, mirroring their roles in the systemic vasculature. Essentially, these hypothesised roles are confirmed, and roles in plexus remodelling, arterial differentiation and perivascular cell recruitment are additionally demonstrated. Extrapolating from the systemic vasculature, such roles are neither unexpected nor novel, however, they have not been previously demonstrated in the coronary vasculature, only indirectly inferred e.g. in recent scRNA-Seq analyses. The work, therefore, offers a sufficient advance to the field and the analysis has been carefully and rigorously performed. Substantive concerns are few:

1) Conceptually, it is difficult to understand why hypoxia should manifest as early as E12.5 in the *Dll4/ Jag1* mutants, if, as inferred, this is due to malformed coronary plexus and perfusion defects. At this stage, SV and endocardial sprouting is not very advanced (in control hearts) and the mutant hearts are less compacted, thus less reliant on vessels for perfusion. Perhaps this could be reconciled if the authors assessed hypoxia in situ e.g. with hypoxyprobe (or at the very least, Glut1 across the whole heart) to correlate regional hypoxia with coronary vessel sprouting. If, instead, this is a stress response to the loss of angiocrine stimuli, as alluded, the authors should validate some putative factors from the RNA-Seq data to support this conclusion.

2) αSMA is used to label smooth muscle cells, however, it is expressed more broadly in mesenchymal cells and cardiomyocytes throughout the embryonic heart prior to ~E16.5 and this is reflected in the images. The authors should attempt to stain for SM22a which, although similarly non-specific, shows better enrichment in VSMCs, relative to the other cell types. Improved visualisation may provide further insight into the nature of the defect e.g. are epicardium-derived precursors trapped or poorly differentiated?

3) How do the authors explain the differences in lethality between the *PdgfbiCr^ERT2^* vs. *Cdh5^CreERT2^* loss of *Dll4*, induced at E12.5?

---

## [Author Response]

Essential revisions:1) Deletion efficiency of genes by in situ or immunostaining detection. The authors claimed that Late endothelial Jag1 or Dll4 inactivation disrupts coronary plexus remodeling. The authors should show the relative target gene Dll4, Jag1 deletion efficiency in inducible mice, by immunostaining or in-situ?

We agree with the reviewer that the efficiency of *Jag1* and *Dll4* deletion in the inducible loss of function models needs to be demonstrated. We now show in the revised Figure 3—figure supplement 1 that *Jag1* and *Dll4* are efficiently deleted after tamoxifen induction of *Pdgfb-iCre^ERT2^*. We have performed ISH for *Jag1* and *Dll4* on *Jag1^flox^;Pdgfb-Cre^ERT^*and *Dll4^flox^;Pdgfb-Cre^ERT2^*hearts that were tamoxifen-induced at E12.5 and E14.5, respectively. As shown in Figure 3—figure supplement 1, *Jag1* or *Dll4* mRNA are severely decreased in coronaries of E15.5 *Jag1^flox^* (or *Dll4^flox^);Pdgfb-cre^ERT2^* mutants. Moreover, we previously showed that *Dll4* mRNA was markedly decreased in E15.5 *Dll4^flox^;Cdh5-Cre^ERT2^* coronaries tamoxifen-induced at E12.5 (see Figure 3—figure supplement 4C).

The revised text says (Results): “Defective coronary remodeling and maturation in endothelial *Jag1* or *Dll4* mutantsWe next examined the requirements of endothelial *Jag1* and *Dll4* for coronary vessel remodeling and maturation. […] This was confirmed by co-immunostaining with SM22a and Notch3 (Figure 3—figure supplement 7C, D) demonstrating near complete absence of smooth muscle differentiation in *Dll4^flox^;Pdgfb-iCre^ERT2^*mutants (Supplementary file 1, sheet 7).”

2) A schematic model need to be provided. Since Jag1 and Dll4 have opposing effects on Notch signaling, based on the current data, how do the authors propose these ligands and Notch signaling shape the pre-artery fate and coronary vessel development temporally and spatially? Since phenotypes were analyzed at different stages during coronary plexus formation, arterial differentiation and myocardial growth, the authors could consider to provide a schematic model to help with the readability.

We thank the reviewers for this suggestion. We have now included a schematic model summarizing our findings on the opposing roles of *Jag1* and *Dll4* during coronary vessel sprouting and their later roles in vessel remodeling. This model is referred to as Figure 8 (new) in the Discussion.

3) More detailed in situ hypoxia measurement need to be included. It is difficult to understand why hypoxia should manifest as early as E12.5 in the Dll4/ Jag1 mutants, if, as inferred, this is due to malformed coronary plexus and perfusion defects. At this stage, SV and endocardial sprouting is not very advanced (in control hearts) and the mutant hearts are less compacted, thus less reliant on vessels for perfusion. Perhaps this could be reconciled if the authors assessed hypoxia in situ e.g. with hypoxyprobe (or at the very least, Glut1 across the whole heart) to correlate regional hypoxia with coronary vessel sprouting. If, instead, this is a stress response to the loss of angiocrine stimuli, as alluded, the authors should validate some putative factors from the RNA-Seq data to support this conclusion.

Following the reviewers suggestion, we have performed immunohistochemical detection of pimonidazole and Glut1 on E12.5 *Jag1^flox^;Nfatc1^Cre^*and *Dll4^flox^;Nfatc1^Cre^* heart sections.

The revised text says (Results): “We examined E12.5 *Jag^1flox^;Nfatc1-Cre* and *Dll4^flox^;Nfatc1-Cre* mutants for evidence of hypoxia given that the gene signatures in the RNA-seq analysis suggested an ongoing hypoxic/metabolic stress response. […] These results indicate that *Jag1^flox^;Nfatc1-Cre* and *Dll4^flox^;Nfatc1-Cre* mutant hearts are not overtly hypoxic at E12.5, suggesting that the hypoxic/metabolic stress gene signatures may be due to a cell autonomous defect of endocardial/endothelial cells”.

Thus, the hydroxyprobe and Glut-1 immunostainings suggests that contrary to our initial interpretation, E12.5 *Jag1^flox^;Nfatc1^Cre^*and *Dll4^flox^;Nfatc1^Cre^* mutants are not more hypoxic than the WT. This would imply that ventricular wall thinning in the mutants is not due to improper coronary plexus development. One possibility is that impaired ventricular wall growth in E12.5 *Jag1^flox^;Nfatc-Cre* and *Dll4^flox^;Nfatc1-Cre* hearts may is due to defective endocardial-myocardial signaling, as we have previously suggested for E10.5 *Dll4^flox^;Nfatc1^Cre^*(D'Amato et al., 2016).

The revised manuscript now says in Discussion: “A common feature among the endocardial or endothelial Notch loss- and gain-of-function models analyzed in our study is the thin ventricular wall (Figure 8B, D). […] This may be clinically relevant to the study and treatment of cardiomyopathies”

Angiocrine factors. Following the reviewer’s suggestion, we also examined the possibility that coronaries might be providing angiocrine factors to sustain myocardial growth and proliferation in absence of coronary perfusion. Tissue-specific instructive functions of ECs have been demonstrated in organ repair without compromising blood supply (Rafii et al., 2016). We have used GSEA (http://software.broadinstitute.org/gsea/index.jsp) to test for the enrichment of a gene set collection of 52 angiocrine factors assembled from Nolan et al. (Nolan et al., 2013) in E12.5 *Jag1^flox^;Nfatc1-Cre* and *Dll4^flox^;Nfatc1-Cre* mutants, relative to controls (see Author response image 1 and Author response table 1). The gene set was enriched with positive normalized enrichment score (NES) in both analyses, indicating that angiocrine factor expression was increased in both mutant conditions (see GSEA profiles). However, enrichment was statistically significant only for the *Jag1^flox^;Nfatc1-Cre* mutant (FDR = 0.04). The leading-edge subset of genes (those contributing mostly to the enrichment score) consisted of 11 genes in both analyses. Seven genes of the 11 genes were shared by both subsets: VEGFA, PDGFB, ANGPT2, EDN1, *FGF9*, CCL9, BMP6 (see heatmaps), suggesting the increased expression of these angiocrine factors occurs independently of the mechanism of angiogenesis in *Jag1^flox^;Nfatc1-Cre* and *Dll4^flox^;Nfatc1-Cre* mutants. We are currently refining our bioinformatics analysis by cross-validating our data with sc-RNA-seq made by (Su et al., 2018). We will validate our data by ISH and eventually, functional analysis, that unfortunately, are out of the scope (and time-limitations) of this manuscript.

**Author response image 1. respfig1:** GSEA analysis of *Jag1^flox^;Nfatc1^Cre^*and *Dll4^flox^;Nfatc1^Cre^* RNA-seq datasets compared against respective controls (Ctrl). Figure 2. GSEA analysis of *Jag1^flox^;Nfatc1^Cre^* and *Dll4^flox^;Nfatc1^Cre^* expression profiles, compared against their respective WT controls. An angiocrine factor gene set consisting of 52 genes was obtained from a previous publication (see Author response table 1 and Nolan et al. Dev Cell. 2013 Jul 29; 26(2)). Of the 52 angiocrine factors, 38 were detected as expressed in the *Dll4* contrast (14,302 genes), and 34 in the *Jag1* contrast (13,982 genes), as represented by the Venn diagrams at the upper left corner (panel A). GSEA results indicated that angiocrine factors were enriched among the genes that are more expressed in both *Jag1* and *Dll4* mutant backgrounds, as indicated by the respective enrichment profiles (panels B and C). In these profiles, red to blue gradient stripes represent the complete collections of expressed genes, sorted by decreasing fold change value in mutant versus WT contrasts: red colours indicate higher expression in mutant background (positive fold change), and blue colours indicate higher expression in WT background (negative fold change). Angiocrine factors, represented by black segments perpendiculary oriented along the gradient stripes, tend to concentrate in positive regions, as indicated by the green enrichment curves. Normalized enrichment score (NES) was therefore positive for both contrasts (1.54 and 0.93, for the *Jag1* and *Dll4* contrasts, respectively), although enrichment was statistically significant only for the *Jag1* analysis (FDR = 0.04). Black rectangles indicate the groups of genes that contribute mostly to calculated NES values, which casually consisted in 11 genes for both analyses. The two leading edge subsets shared 7 genes (VEGFA, PDGFB, ANGPT2, EDN1, *FGF9*, CCL9 and BMP6), as represented by the Venn diagrams at the lower left corner (panel F). The central scatterplot (panel D) compares the fold change rank of the set of 34 angiocrine factors that were detected in both contrasts, and indicates (with a colour code) which of them were detected as differentially expressed (with p-value < 0.05) in each of the contrasts. Six leading edge subset angiocrine factors were detected as differentially expressed in at least one contrast, and four of them (VEGFA, BMP6, EDN1 and PDGFB) were detected in both. Heatmaps represent relative expression values for 38 angiocrine factors in *Dll4* KO samples and their respective WT controls (panel E), and 34 angiocrine factors in *Jag1* KO samples and the respective WT controls (panel G). Red and blue colours represent expression values above and below average, respectively. The four genes with asterisks in panel E are those that were detected as expressed only in the DLL4 contrast.

**Author response table 1. resptable1:** List of angiocrine gene from Nolan et al. (2013) compared with our RNA-seq data from *Jag1^flox^;Nfatc1-Cre* and *Dll4^flox^;Nfatc1-Cre* embryos (Figure 2A and Supplementary file 3).

Nolan (52 genes)	*Dll4* (38 genes)	*Jag1* (34 genes)	*Dll4* and *Jag1* (34 genes)	Not common to *Dll4* and *Jag1* (14 genes)	*Dll4* specific (4 genes)
*ANGPT2*	*ANGPT2*	*ANGPT2*	*ANGPT2*	*CCL3*	*CCL6*
*BMP2*	*BMP2*	*BMP2*	*BMP2*	*CXCL2*	*CXCL13*
*BMP5*	*BMP5*	*BMP5*	*BMP5*	*DKKL1*	*FGF7*
*BMP6*	*BMP6*	*BMP6*	*BMP6*	*GJB2*	*IL1F9*
*CCL2*	*CCL2*	*CCL2*	*CCL2*	*IL1A*	
*CCL3*	*CCL6*	*CCL9*	*CCL9*	*IL6*	
*CCL6*	*CCL9*	*COL14A1*	*COL14A1*	*LAMA-4*	
*CCL9*	*COL14A1*	*COL4A1*	*COL4A1*	*LAMB1-1*	
*COL14A1*	*COL4A1*	*CXCL12*	*CXCL12*	*MMP13*	
*COL4A1*	*CXCL12*	*DKK2*	*DKK2*	*MMP27*	
*CXCL12*	*CXCL13*	*EDN1*	*EDN1*	*MMP8*	
*CXCL13*	*DKK2*	*EGFL7*	*EGFL7*	*MMP9*	
*CXCL2*	*EDN1*	*ELN*	*ELN*	*TNF*	
*DKK2*	*EGFL7*	*ESM1*	*ESM1*	*WNT8A*	
*DKKL1*	*ELN*	*FGF1*	*FGF1*		
*EDN1*	*ESM1*	*FGF9*	*FGF9*		
*EGFL7*	*FGF1*	*FN1*	*FN1*		
*ELN*	*FGF7*	*GJA1*	*GJA1*		
*ESM1*	*FGF9*	*GJA4*	*GJA4*		
*FGF1*	*FN1*	*GJC1*	*GJC1*		
*FGF7*	*GJA1*	*IGF1*	*IGF1*		
*FGF9*	*GJA4*	*IL33*	*IL33*		
*FN1*	*GJC1*	*JAG1*	*JAG1*		
*GJA1*	*IGF1*	*KITL*	*KITL*		
*GJA4*	*IL1F9*	*LAMB2*	*LAMB2*		
*GJB2*	*IL33*	*LAMC1*	*LAMC1*		
*GJC1*	*JAG1*	*PDGFB*	*PDGFB*		
*IGF1*	*KITL*	*PDGFC*	*PDGFC*		
*IL1A*	*LAMB2*	*PDGFD*	*PDGFD*		
*IL1F9*	*LAMC1*	*TGFB2*	*TGFB2*		
*IL33*	*PDGFB*	*VEGFA*	*VEGFA*		
*IL6*	*PDGFC*	*VEGFC*	*VEGFC*		
*JAG1*	*PDGFD*	*WNT2*	*WNT2*		
*KITL*	*TGFB2*	*WNT5A*	*WNT5A*		
*LAMA-4*	*VEGFA*				
*LAMB1-1*	*VEGFC*				
*LAMB2*	*WNT2*				
*LAMC1*	*WNT5A*				
*MMP13*					
*MMP27*					
*MMP8*					
*MMP9*					
*PDGFB*					
*PDGFC*					
*PDGFD*					
*TGFB2*					
*TNF*					
*VEGFA*					
*VEGFC*					
*WNT2*					
*WNT5A*					
*WNT8A*					

4) Other more specific markers for smooth muscle should be used. αSMA is used to label smooth muscle cells, however, it is expressed more broadly in mesenchymal cells and cardiomyocytes throughout the embryonic heart prior to ~E16.5 and this is reflected in the images. The authors should attempt to stain for SM22a which, although similarly non-specific, shows better enrichment in VSMCs, relative to the other cell types. Improved visualization may provide further insight into the nature of the defect e.g. are epicardium-derived precursors trapped or poorly differentiated?

In our study we used double staining of pericytes and smooth muscle using Notch3 and αSMA respectively. This method was used previously by (Volz et al., 2015), to demonstrate that pericytes are precursors of smooth muscle cells in coronaries in E14.5-E16.5 hearts. We cite this paper (Volz et al., 2015). Moreover, we avoided αSMA-immunostained mesenchyme and cardiomyocytes by counting αSMA/Notch3 double-positive cells surrounding coronaries, thus we are sure that the cell type that we label with αSMA is vascular smooth muscle. Nevertheless, to address the reviewer´s concern we have performed Notch3/SM22a double immunostaining on *Pdgfb-iCre^ERT2/+^;Jag1^flox/flox^*(and *Dll4^flox/flox^)* heart sections in triplicate. After quantifying the percentage of SM22a/Notch3+-positive staining in Notch3-positive cells, we obtained very similar results, namely that smooth muscle differentiation is significantly impaired in *Pdgfb-iCre^ERT2/+^;Jag1^flox/flox^*(and *Dll4^flox/flox^)* mutants (new Figure 3—figure supplement 2A). These data indicate that *Jag1* and *Dll4* are required for coronary vascular smooth muscle cell differentiation.

The revised text says (Results): “Although αSMA is a commonly used marker of vascular smooth muscle cells, it is expressed more broadly in mesenchymal cells and cardiomyocytes throughout the embryonic heart prior to E16.5. […] After co-staining with Notch3, we found that the proportion of SM22a and Notch3-positive cells in the *Jag1^flox^;Pdgfb-iCre^ERT2^*mutants was significantly reduced (Supplementary file 1, sheet 7), confirming that pericytes fail to properly differentiate into smooth muscle.”

“Furthermore, *Dll4^flox^;Pdgfb-iCre^ERT2^*mutants had deficient perivascular cell coverage as indicated by decreased αSMA- and Notch3-positive cells (Supplementary file 1, sheet 6). This was confirmed by co-immunostaining with SM22a and Notch3 (Figure 3—figure supplement 7C, D) demonstrating near complete absence of smooth muscle differentiation in *Dll4^flox^;Pdgfb-iCre^ERT2^*mutants (Supplementary file 1, sheet 7)”.

Reference:

Nolan DJ, Ginsberg M, Israely E, Palikuqi B, Poulos MG, James D, Ding BS, Schachterle W, Liu Y, Rosenwaks Z et al. 2013. Molecular signatures of tissue-specific microvascular endothelial cell heterogeneity in organ maintenance and regeneration. *Dev Cell***26**: 204-219.